# LayoutGPT: Compositional Visual Planning and Generation with Large Language Models

**Weixi Feng**[1]*  **Wanrong Zhu**[1]*  **Tsu-jui Fu**[1]  **Varun Jampani**[2]  **Arjun Akula**[2]
**Xuehai He**[3]  **Sugato Basu**[2]  **Xin Eric Wang**[3]  **William Yang Wang**[1]

[1]University of California, Santa Barbara
[2]Google
[3]University of California, Santa Cruz

https://github.com/weixi-feng/LayoutGPT

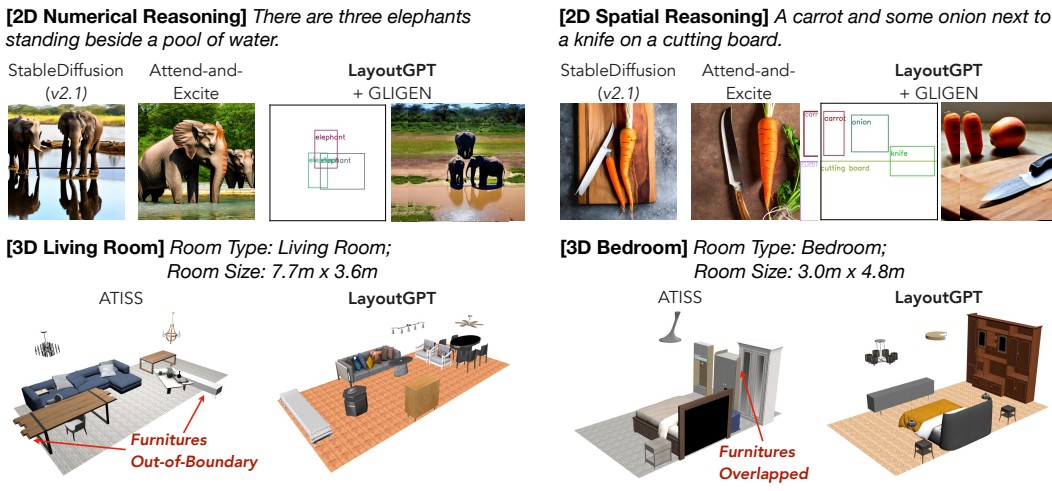

Figure 1: Generated layouts from LayoutGPT in 2D images and 3D indoor scenes. LayoutGPT can serve as a visual planner to reflect challenging numerical and spatial concepts in visual spaces.

## Abstract

Attaining a high degree of user controllability in visual generation often requires intricate, fine-grained inputs like layouts. However, such inputs impose a substantial burden on users when compared to simple text inputs. To address the issue, we study how Large Language Models (LLMs) can serve as visual planners by generating layouts from text conditions, and thus collaborate with visual generative models. We propose LayoutGPT, a method to compose in-context visual demonstrations in style sheet language to enhance the visual planning skills of LLMs. LayoutGPT can generate plausible layouts in multiple domains, ranging from 2D images to 3D indoor scenes. LayoutGPT also shows superior performance in converting challenging language concepts like numerical and spatial relations to layout arrangements for faithful text-to-image generation. When combined with a downstream image generation model, LayoutGPT outperforms text-to-image models/systems by 20-40% and achieves comparable performance as human users in designing visual layouts for numerical and spatial correctness. Lastly, Layout-GPT achieves comparable performance to supervised methods in 3D indoor scene synthesis, demonstrating its effectiveness and potential in multiple visual domains.

---

*equal contribution, correspondence to {weixifeng, wanrongzhu}@cs.ucsb.edu

37th Conference on Neural Information Processing Systems (NeurIPS 2023).

# 1 Introduction

Can Large Language Models (LLMs) comprehend visual concepts and generate plausible arrangments in visual spaces? Recently, LLMs have shown significant advancement in various reasoning skills [50, 49] that remain challenging to visual generative models. For instance, text-to-image generation (T2I) models suffer from generating objects with specified counts, positions, and attributes [10, 24]. 3D scene synthesis models face challenges in preserving furniture within pre-defined room sizes [30]. Addressing these issues necessitates the development of compositional skills that effectively arrange components in a coherent manner, accurately reflecting object specifications and interactions.

Visual layout is an essential symbolic representation that has been widely studied as it reflects the compositions of a visual space [33, 53, 45, 34]. For instance, layout generation models [21, 25, 17, 53, 23] can be combined with region-controlled image generation methods [56, 27] to improve image compositionality [52]. But unlike LLMs, these models are restricted to discrete categories or have limited reasoning skills for complicated text conditions. Recently, LLMs like ChatGPT [37], are adopted as a centralized module of frameworks or systems where multiple foundational computer vision models are integrated. Through defined action items or API calls, LLMs can interact with visual generative models to extend the systems' capability into image generation tasks. [51].

Despite the advancement, existing approaches that involve the collaboration between LLMs and image generation models are either limited to executing the latter through program generation or using LLMs for language data augmentation for image editing [3]. Current LLM-based systems fail to improve the compositional faithfulness of a generated image by simply using T2I models through API calls. While one could additionally integrate models that synthesize images with the guidance of layouts [56, 27], keypoints [27], or sketches [20, 57], users still have to create fine-grained inputs on their own, leading to extra efforts and degraded efficiency compared to pure language instructions.

To address these challenges, we introduce **LayoutGPT**, a training-free approach that injects visual commonsense into LLMs and enables them to generate desirable layouts based on text conditions. Despite being trained without any image data, LLMs can learn visual commonsense through in-context demonstrations and then apply the knowledge to infer visual planning for novel samples. Specifically, we observe that representing image layouts is highly compatible with how style sheet language formats images on a webpage. Therefore, as LLMs are trained with program data, constructing layouts as structured programs may enhance LLMs' ability to "imagine" object locations from merely language tokens. Our programs not only enable stable and consistent output structures but also strengthen LLMs' understanding of the visual concepts behind each individual attribute value. When combined with a region-controlled image generation model [27], LayoutGPT outperforms existing methods by 20-40% and achieves comparable performance as human users in generating plausible image layouts and obtaining images with the correct object counts or spatial relations.

In addition, we extend LayoutGPT from 2D layout planning to 3D indoor scene synthesis. With a slight expansion of the style attributes, LayoutGPT can understand challenging 3D concepts such as depth, furniture sizes, and practical and coherent furniture arrangements for different types of rooms. We show that LayoutGPT performs comparably to a state-of-the-art (SOTA) supervised method. Our experimental results suggest that LLMs have the potential to handle more complicated visual inputs. Our contribution can be summarized as the following points:

- We propose LayoutGPT, a program-guided method to adopt LLMs for layout-based visual planning in multiple domains. LayoutGPT addresses the *inherent* multimodal reasoning skills of LLMs and can improve end-user efficiency.

- We propose **N**umerical and **S**patial **R**easoning (NSR-1K) benchmark that includes prompts characterizing counting and positional relations for text-to-image generation.

- Experimental results show that LayoutGPT effectively improves counting and spatial relations faithfulness in 2D image generation and achieves strong performance in 3D indoor scene synthesis. Our experiments suggest that the reasoning power of LLMs can be leveraged for visual generation and handling more complicated visual representations.

## 2 Related Work

**Image Layout Generation** Layout generation has been an important task for automatic graphical design for various scenarios, including indoor scenes [40, 46], document layouts [59, 60, 15], and graphical user interface [8]. Previous work has proposed various types of models that need to be trained from scratch before generating layouts. LayoutGAN [25] is a GAN-based framework to generate both class and geometric labels of wireframe boxes for a fixed number of scene elements. LayoutVAE [21] generates image layouts conditioned on an input object label set. Transformer-based methods are proposed to enhance flexibility in the layout generation process. For instance, LayoutTransformer [17] adopts self-attention to learn contextual relations between elements and achieve layout completion based on a partial layout input. BLT [23] proposes a hierarchical sampling policy so that any coordinate values can be modified at the sampling stage to enable flexible and controlled generation. However, existing methods are restricted to class labels and fail to reason over numerical and spatial concepts in text conditions. In contrast, LayoutGPT can convert challenging textual concepts to 2D layouts and generate free-form, detailed descriptions for each region.

**Compositional Image Generation** Recent studies have shown that text-to-image generation (T2I) models suffer from compositional issues such as missing objects, incorrect spatial relations, and incorrect attributes [24, 2]. StructureDiffusion [10] proposes to adjust text embeddings by utilizing prior knowledge from linguistic structures. Attend-and-Excite [4] optimizes attention regions so that objects attend on separate regions. Another line of work strives to introduce extra conditions as inputs. For example, ReCo [56], GLIGEN [27], and Layout-Guidance [6] can generate images based on bounding box inputs and regional captions. [52] combines a layout generator and a region-controlled method to achieve accurate generation results. While we focus on layout generation, we also employ layout-to-image models to generate final images and show the effectiveness of LayoutGPT.

**Indoor Scene Synthesis** Indoor scene synthesis aims at generating reasonable furniture layouts in a 3D space that satisfies room functionality. Early work adopting autoregressive models requires supervision of 2D bounding boxes and other visual maps [40]. Later, SceneFormer [47] proposes to apply a set of transformers to add furniture to scenes. While previous work adopts separate models to predict different object attributes, ATISS [38] demonstrates that a single transformer model can generate more realistic arrangments while being more efficient. In this work, we investigate leveraging LLMs to achieve scene synthesis without any fine-tuning.

**LLMs for Vision** Language inputs have been an essential part of many vision language tasks [43, 11, 28, 14]. With the strong generalization ability of contemporary LLMs, recent work attempts to adapt the power of LLMs on multimodal tasks [31, 55]. For instance, multimodal chain-of-thought [58] trained a model to incorporate visual inputs as rationales for question answering. [22] proposes to learn translation parameters to map embeddings between visual and language domains such that an LLM can ground on both modalities. VisProg [18] and ViperGPT [44] use LLMs to design modular pseudocode instructions or executable Python programs to achieve visual reasoning. LLMScore [32] leverages LLMs to evaluate text-to-image models. Visual ChatGPT [51] proposes a prompt manager that supports the execution of various image generation models. In this work, we directly involve LLMs in the generation process by leveraging LLMs to design visual layouts through in-context learning and structured representations.

## 3 Method

### 3.1 Overview

Given a condition $\mathcal{C}$, the goal of layout generation is to predict a set of tuples $\mathcal{O} = \{\mathbf{o}_j | j = 1, 2, \ldots, n\}$ where each tuple $\mathbf{o}_j$ denotes the layout information of a 2D or 3D bounding box of object $j$. In image planning, $\mathcal{C}$ is the input text prompt, $\mathbf{o}_j$ consists of a category $c_j$, bounding box location $\mathbf{t}_j = (x_j, y_j) \in \mathbb{R}^2$ and bounding box size $\mathbf{s}_j = (w_j, h_j) \in \mathbb{R}^2$, i.e. $\mathbf{o}_j = (c_j, \mathbf{t}_j, \mathbf{s}_j)$. Similarly, in 3D scene synthesis, $\mathcal{C}$ specifies the room type and room size, $\mathbf{o}_j$ consists of category $c_j$, location $\mathbf{t}_j \in \mathbb{R}^3$, size $\mathbf{s}_j \in \mathbb{R}^3$, and orientation $\mathbf{r}_j \in \mathbb{R}$, i.e. $\mathbf{o}_j = (c_j, \mathbf{t}_j, \mathbf{s}_j, \mathbf{r}_j)$. While $c_j$ can be modeled as a discrete value, our method directly predicts the category text.

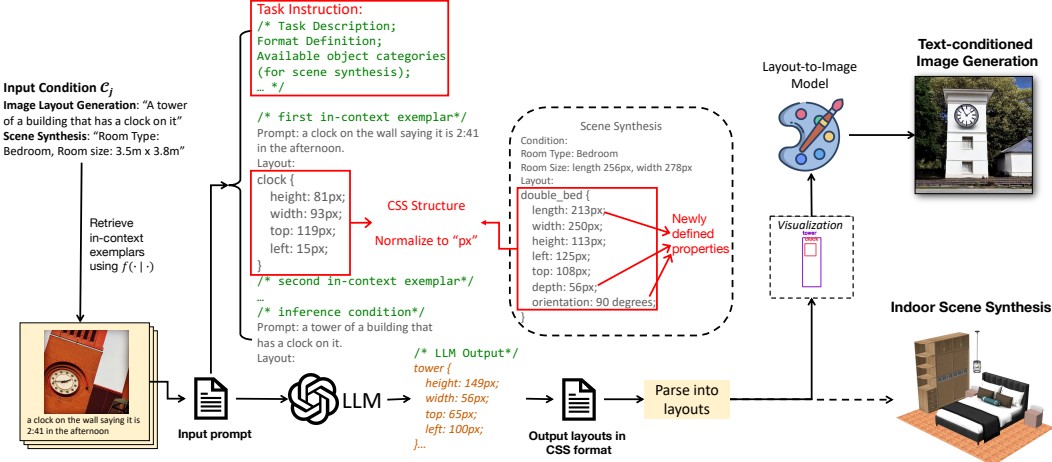

Figure 2: The overview process of our LayoutGPT framework performing 2D layout planning for text-conditioned image generation or 3D layout planning for scene synthesis.

## 3.2 LayoutGPT Prompt Construction

As is shown in Fig. 2, LayoutGPT prompts consist of three main components: **task instructions**, and in-context exemplars in **CSS structures** with **normalization**.

**CSS Structures** In autoregressive layout generation, $\mathbf{o}_j$ is usually modeled as a plain sequence of values, i.e. $(c_1, x_1, y_1, w_1, h_1, c_2, x_2, \ldots)$ [17, 23]. However, such a sequence can be challenging for LLMs to understand due to underspecified meaning of each value. Therefore, we seek a structured format that specifies the physical meaning of each value for LLMs to interpret spatial knowledge. We realize that image layouts are highly similar to how CSS (short for Cascading Style Sheets) formats the layout of a webpage and defines various properties of the `img` tag in HTML. For instance, $x_j, y_j$ corresponds to the standard properties `left` and `top`, while $w_j, h_j$ corresponds to `width` and `height` in CSS. As LLMs like GPT-3.5/4 are trained with code snippets, formatting image/scene layouts in CSS structures potentially enhances the LLMs' interpretation of the spatial meaning behind each value. Therefore, as is shown in Fig. 2, we place category name $c_j$ as the selector and map other attribute values into the declaration section following standard CSS styles.

**Task Instructions & Normalization** Similar to previous work in improving the prompting ability of LLMs [48, 42, 37], we prepend task instructions to the prompt to specify the task goal, define the standard format, unit for values, etc. Besides, as the common length unit of CSS is pixels (px), we normalize each property value based on a fixed scalar and rescale the value to a maximum of 256px. As will be shown in later sections (Sec. 4.4 & 5.4), all three components play important roles in injecting visual commonsense into LLMs and improving generation accuracy.

## 3.3 In-Context Exemplars Selection

Following previous work [1, 54], we select supporting demonstration exemplars for in-context learning based on retrieval results. Given a test condition $\mathcal{C}_j$ and a support set of demonstrations $\mathcal{D} = \{(\mathcal{C}_k, \mathbf{o}_k)|k = 1, 2, \ldots\}$, we define a function $f(\mathcal{C}_k, \mathcal{C}_j) \in \mathbb{R}$ that measures the distances between two conditions. For 2D text-conditioned image layout generation, we adopt the CLIP [39] model to extract text features of $\mathcal{C}_j$ (usually a caption) and the image feature of $\mathcal{C}_k$ and measure the cosine similarity between them. For the 3D scene synthesis task where each room has length $rl$ and width $rw$, we measure distance with $f(\mathcal{C}_k, \mathcal{C}_j) = \|rl_k - rl_j\|^2 + \|rw_k - rw_j\|^2$. We select supporting demonstrations with the top-$k$ least distance measures and construct them as exemplars following the CSS structure in Fig. 2. These supporting examples are provided to GPT-3.5/4 in reverse order, with the most similar example presented last.

Table 1: Dataset statistics and examples of the NSR-1K benchmark for image layout planning and text-to-image (T2I) generation with an emphasis on numerical and spatial reasoning.

| Task | Type | Example Prompt | # Train | # Val | # Test |
|------|------|----------------|---------|-------|--------|
| T2I Numerical Reasoning | Single Category | *"There are two giraffes in the photo."* | 14890 | - | 114 |
| | Two Categories | *"Three potted plants with one vase in the picture."* | 7402 | - | 197 |
| | Comparison | *"A picture of three cars with a few fire hydrants, the number of cars is more than that of fire hydrants."* | 7402 | - | 100 |
| | Natural | *"A fenced in pasture with four horses standing around eating grass."* | 9004 | - | 351 |
| T2I Spatial Reasoning | Two Categories | *"A dog to the right of a bench."* | 360 | - | 199 |
| | Natural | *"A black cat laying on top of a bed next to pillows."* | 378 | | 84 |

## 3.4 Image and Scene Generation

For text-conditioned image synthesis, we utilize a layout-to-image generation model to generate images based on the generated layouts. As for each object layout in 3D scene synthesis, we retrieve a 3D object based on the predicted category, location, orientation, and size following [38]. We directly render the scene with the retrieved 3D objects. See Sec. 4 & Sec. 5 for more details.

# 4 LayoutGPT for Text-Conditioned Image Synthesis

In this section, we provide an extensive evaluation of LayoutGPT for 2D text-to-image (T2I) synthesis and compare it with SOTA T2I models/systems. An ablation study is conducted to demonstrate the effect of individual components from LayoutGPT. We also showcase qualitative results and application scenarios of our method.

## 4.1 Experiment Setup

**Datasets & Benchmarks** To evaluate the generations in terms of specified counts and spatial locations, we propose NSR-1K, a benchmark that includes template-based and human-written (natural) prompts from MSCOCO [29]. Table 1 summarizes our dataset statistics with examples. For template-based prompts, we apply a set of filters to obtain images with only 1-2 types of object and then create prompts based on object categories and bounding box information. As for natural prompts, we extract COCO captions with keywords to suit the task of numerical reasoning (e.g. "four") or spatial reasoning (e.g. "on top of") and ensure that all objects from the bounding box annotations are mentioned in the caption to avoid hallucination. Each prompt from NSR-1K is guaranteed to have a corresponding ground truth image and layout annotations. Detailed benchmark construction processes are described in Appendix B.1.

**Evaluation Metrics** To evaluate generated layouts, we report precision, recall, and accuracy based on generated bounding box counts and spatial positions [9, 16]. For spatial reasoning, each prompt falls into one of the four types of relations ({*left, right, top, below*}) and we use the bounding box center for evaluation following PaintSkills [7]. To evaluate generated images, we first obtain bounding boxes from GLIP [26] detection results and then compute average accuracy based on the bounding box counts or spatial relations. We also report CLIP cosine similarity between text prompts and generated images for reference. Detailed metric descriptions are listed in Appendix B.2.

**Baselines** As we consider both layout evaluation and image evaluation, we compare LayoutGPT with **end-to-end T2I models** (Stable Diffusion [41], Attend-and-Excite [4])[2] and **two-stage systems** that generate layouts first and then apply GLIGEN [27] as the layout-to-image model. We also evaluate ground truth layouts and human-drawn layouts as the theoretical upper bounds. The human-drawn layouts are collected through crowdsourcing, in which we specifically ask human annotators to draw layouts given text prompts. We slightly modify LayoutTransformer [17] as a baseline for supervised conditional layout generation. Detailed descriptions of baseline setups and human annotating are discussed in the Appendix A and E.

---

[2]Attend-and-Excite uses Stable Diffusion (SD) as the generative backbone. For both end-to-end T2I models, we report results on SD v1.4 and SD v2.1.

Table 2: Comparison of our LayoutGPT with baseline methods in terms of counting and spatial correctness. Line 5-11 generates layout and adopts GLIGEN [27] for layout-guided image generation. "Human" (line 11) denotes layouts collected from human users given text prompts. Text in bold shows the best results of LayoutGPT.

| | | Numerical Reasoning | | | | | Spatial Reasoning | | |
| | | Layout Eval. | | | Image Eval. | | Layout Eval. | Image Eval. | |
| | Methods | Precision | Recall | Accuracy | Acc. (GLIP) | CLIP Sim. | Accuracy | Acc. (GLIP) | CLIP Sim. |
|---|---|---|---|---|---|---|---|---|---|
| | *Text ⟶ Image* | | | | | | | | |
| 1 | Stable Diffusion (v1.4) [41] | - | - | - | 32.22 | 0.256 | - | 16.89 | 0.252 |
| 2 | Stable Diffusion (v2.1) | - | - | - | 42.44 | 0.256 | - | 17.81 | 0.256 |
| 3 | Attend-and-Excite (SD v1.4) [4] | - | - | - | 38.96 | 0.258 | - | 24.38 | 0.263 |
| 4 | Attend-and-Excite (SD v2.1) | - | - | - | 45.74 | 0.254 | - | 26.86 | 0.264 |
| | *Text → Layout → Image* | | | | | | | | |
| 5 | LayoutTransformer [17] | 75.70 | 61.69 | 22.26 | 40.55 | 0.247 | 6.36 | 28.13 | 0.241 |
| 6 | LayoutGPT (GPT-3.5) | **94.81** | **96.49** | **86.33** | 51.20 | 0.258 | 82.54 | 52.86 | 0.264 |
| 7 | LayoutGPT (Codex) | 90.19 | 88.29 | 72.02 | 46.64 | 0.254 | 74.63 | 45.58 | 0.262 |
| 8 | LayoutGPT (GPT-3.5, chat) | 81.84 | 85.47 | 75.51 | 54.40 | **0.261** | 85.87 | 56.75 | **0.268** |
| 9 | LayoutGPT (GPT-4) | 78.36 | 86.29 | 78.43 | **55.64** | **0.261** | **91.73** | **60.64** | **0.268** |
| 10 | GT layouts | 100.00 | 100.00 | 100.00 | 53.23 | 0.256 | 100.00 | 62.54 | 0.261 |
| 11 | Human | 99.26 | 96.52 | 92.56 | 56.07 | 0.258 | 91.17 | 51.94 | 0.258 |

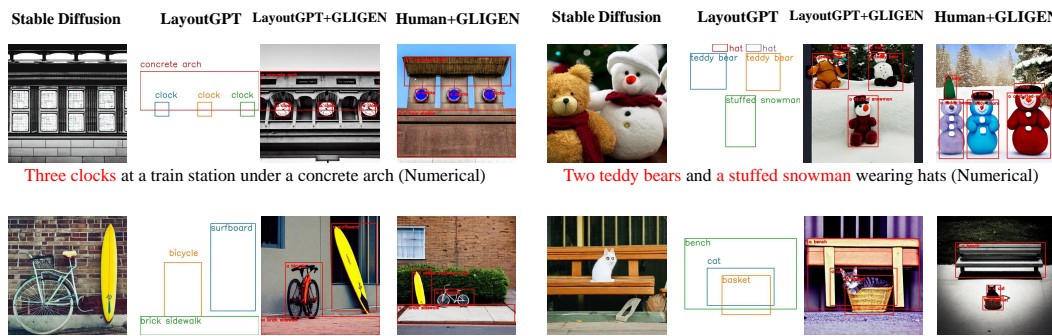

Three clocks at a train station under a concrete arch (Numerical)    Two teddy bears and a stuffed snowman wearing hats (Numerical)

A yellow surfboard sits next to a bicycle on a brick sidewalk (Spatial)    A cat is sitting on a basket under a bench (Spatial)

Figure 3: Qualitative comparison between Stable Diffusion, LayoutGPT, and human annotations regarding numerical (top row) and spatial reasoning (bottom row) skills.

## 4.2  Evaluation Results

**Quantitative Results** As shown in Table 2, among the variants of LayoutGPT (#6-#9), GPT-3.5 achieves the best performance in numerical reasoning while GPT-4 performs the best in generating correct spatial positions. LayoutGPT outperforms LayoutTransformer (#5) by large margins, proving the strong cross-modal reasoning skills of LLMs. As for image-level evaluation, LayoutGPT surpasses end-to-end T2I models (#1-#3) by 20-40% in GLIP-based accuracy and relatively 1-6% in CLIP similarity. Therefore, using layouts as an intermediate representation indeed leads to more reliable and faithful generation outcomes. In addition, LayoutGPT achieves similar layout accuracy as human users (numerical #6 vs. #11 (86.33% v.s. 92.56%); spatial #9 vs. #11 (91.73% v.s. 91.17%)), which implies its potential to spare users from drawing layouts manually. The discrepancy between layout accuracy and GLIP-based accuracy suggests that the bottleneck mainly stems from layout-guided image generation and GLIP grounding results.

In addition, LayoutGPT binds attributes to each object's bounding box with 100% accuracy on HRS [2] color prompts. We further evaluate the attribute correctness rate (accuracy) on the final generated images when combining LayoutGPT with GLIGEN/ReCo. As shown in Table 3, our system largely improves the color correctness over Stable Diffusion with multiple objects.

**Qualitative results** We show the qualitative results of LayoutGPT and baselines in Fig. 3. LayoutGPT can understand visual commonsense such as the clock sizes at a train station (top left) or complex spatial relations between multiple objects (bottom right), while SD fails to generate correct numbers or positions. Besides, LayoutGPT demonstrates a similar layout design to human users (bottom left). Fig. 11 in the Appendix visualizes the results of attribute binding using LayoutGPT and ReCo [56].

Table 3: Color binding accuracy evaluated on prompts from HRS-Bench [2]. We follow the benchmark and use a hue-based classifier to identify the color of generated objects.

| Models | Attribute binding Accuracy (%) | | | |
|---|---|---|---|---|
| | Prompts w/ 2 objects | Prompts w/ 3 objects | Prompts w/ 4 objects | Overall |
| SD1.4 | 18.57 | 10.10 | 11.36 | 12.84 |
| Attend-and-Excite | 31.43 | 19.19 | 20.45 | 22.96 |
| LayoutGPT + GLIGEN | 22.86 | 19.19 | 14.77 | 18.68 |
| LayoutGPT + ReCo [56] | **40.00** | **37.37** | **34.09** | **36.96** |

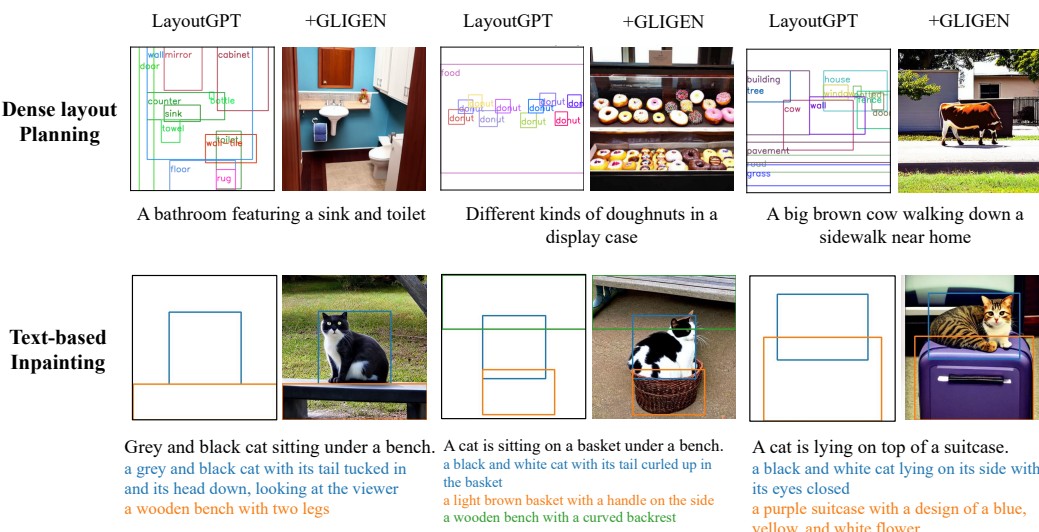

Figure 4: **Dense layout planning**: LayoutGPT can generate rich objects or categories in complex scenes for MSCOCO 2017 Panoptic prompts [29]. **Text-based inpainting**: LayoutGPT can generate free-form regional descriptions that are not mentioned in the global prompt.

## 4.3 Application Scenarios

By utilizing LLMs as layout generators, LayoutGPT can be applied to a diverse set of scenarios for accurate and creative image generation.

**Dense Layout Planning**: In Fig. 4 (top), we apply random in-context examples from COCO17 panoptic annotations with 6∼15 bounding boxes per image. LayoutGPT can be applied to scenarios that imply numerous objects (e.g. different kinds of donuts) or various categories (e.g. bathroom or street view). Though only a few objects are mentioned in the prompts, LayoutGPT predicts layouts for the whole scene and imagines common objects that are usually visible in each scene.

**Text-based Inpainting**: In addition, the inherent language generation ability of LLMs enables our method to generate fine-grained regional descriptions from coarse global prompts (Fig. 4 bottom). LayoutGPT can enrich the description of each object with details that are not mentioned in the prompt, producing suitable outputs for models like ReCo [56].

**Counterfactual Scenarios**: We test LayoutGPT on counterfactual prompts provided by GPT-4 [35]. The in-context examples are randomly drawn from MSCOCO 2017[29], which greatly differs from the counterfactual prompts. As shown in Fig. 5, LayoutGPT manages to generate reasonable layouts on these challenging prompts and handles the relationship between objects well.

## 4.4 Ablation Study

**Component Analysis** Table 4 presents the component analysis of our CSS-style prompt on spatial reasoning prompts. Comparisons between line 1-3 entails that the task instructions (#2) and CSS format (#3) effectively improve layout accuracy. Format in-context exemplars in CSS structures

Table 4: Ablation study of LayoutGPT (GPT-3.5) on spatial reasoning prompts. "w/ Instr.": with prepended task instructions. "w/ CSS": format in-context demonstrations in CSS style. "w/ Norm.": normalizing attribute values to integers by a fixed size.

| | w/ Instr. | w/ CSS | w/ Norm. | Layout-to-Image Model | Layout Eval | Image Eval | |
|---|---|---|---|---|---|---|---|
| | | | | | Acc. | Acc. (GLIP) | CLIP Sim |
| 1 | | | | | 55.12 | 34.35 | 0.259 |
| 2 | ✓ | | | | 78.23 | 47.92 | 0.263 |
| 3 | | ✓ | | | 80.82 | 51.38 | 0.264 |
| 4 | | | ✓ | | 44.10 | 26.43 | 0.257 |
| 5 | ✓ | ✓ | | GLIGEN [27] | 81.84 | 52.08 | 0.264 |
| 6 | ✓ | | ✓ | | 73.36 | 44.88 | 0.262 |
| 7 | | ✓ | ✓ | | 76.61 | 47.56 | 0.263 |
| 8 | ✓ | ✓ | ✓ | | **82.54** | **52.86** | **0.264** |
| 9 | ✓ | ✓ | ✓ | Layout-Guidance [6] | 82.54 | 31.02 | 0.258 |
| 10 | GT layouts | | | | 100.00 | 33.92 | 0.257 |

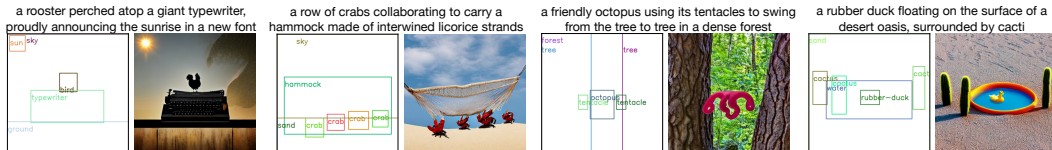

Figure 5: Qualitative examples of LayoutGPT's performance on counterfactual prompts.

show a more significant effect on accuracy. Pairwise comparisons of line `5-7` support the argument that the CSS style is the most essential component. While solely applying normalization degrades accuracy in line `4`, line `5&8` shows that it slightly improves the performance when combined with other components.

**Model-Agnostic Property** We show that LayoutGPT is agnostic to layout-guided image generation models in line `9-10` in Table 4. We feed the same generated layouts from LayoutGPT to Layout-Guidance [6] and compute image-level metrics. Compared to using ground truth layouts (`#10`), LayoutGPT (`#9`) shows a minor gap in GLIP-based accuracy and a comparable CLIP similarity score. The discrepancy in GLIP-based accuracy is similar to that in Table 2, implying that the layouts generated by our method are agnostic to the downstream model.

## 5 LayoutGPT for Indoor Scene Synthesis

### 5.1 Task Setup

**Datasets & Benchmarks** For indoor scene synthesis, we use an updated version of the 3D-FRONT dataset [12, 13] following ATISS [38]. After applying the same pre-processing operations, we end up with 4273 bedroom scenes and 841 scenes for the living room. We only use rectangular floor plans of the test set for evaluation since LayoutGPT is not compatible with irregular ones. Hence, we end up with 3397/453/423 for train/val/test split of bedroom scenes and 690/98/53 for train/val/test split of living room scenes.

**Evaluation Metrics** We follow prior work [38] to report KL divergence between the furniture category distributions of predicted and ground truth scenes. We also render scene images from four camera angles for each scene and report FID scores [19]. In addition, we report out-of-bound rates, i.e. the percentage of scenes with furniture exceeding the floor plan boundary.

### 5.2 Evaluation Results

**Quantitative Results** The evaluation results are recorded in Table 5. We provide a random baseline for comparison denoted as "Random Scenes", in which the scene is randomly sampled from the in-context exemplars for each inference run.[3]

---

[3]Notice that while the scenes in "Random Scenes" are sampled from the training set, the out-of-boundary rate is larger than 0 since the 3D-FRONT dataset contains a small portion of scenes with out-of-bound furniture.

Table 5: Comparison of LayoutGPT with ATISS on indoor scene synthesis. "Random Scenes" means randomly sampling one training scene from the in-context demonstrations for each inference room sample. (* denotes results reproduced by us)

| Models | Bedrooms | | | Living Rooms | | |
|---|---|---|---|---|---|---|
| | Out of bounds ($\downarrow$) | KL Div. ($\downarrow$) | FID ($\downarrow$) | Out of bounds ($\downarrow$) | KL Div. ($\downarrow$) | FID ($\downarrow$) |
| Random Scenes | 11.16 | 0.0142 | 23.76 | 9.43 | 0.1239 | 79.61 |
| ATISS*[17] | 49.88 | **0.0113** | 30.02 | 83.02 | **0.1054** | 85.40 |
| LayoutGPT (GPT-3.5) | **43.26** | 0.0995 | **28.37** | 73.58 | 0.1405 | **76.34** |
| LayoutGPT (GPT-3.5, chat) | 57.21 | 0.0846 | 29.66 | 81.13 | 0.2077 | 89.40 |
| LayoutGPT (GPT-4) | 51.06 | 0.1417 | 29.88 | **64.15** | 0.1613 | 78.60 |

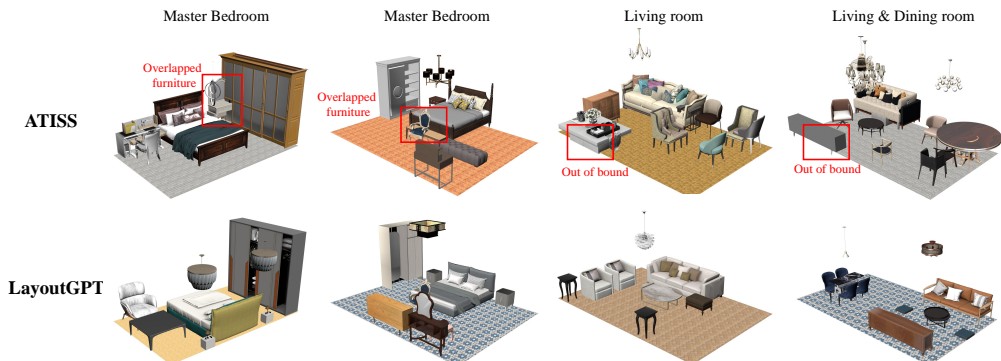

Figure 6: Visualization of LayoutGPT across different types of rooms with different floor plan sizes.

For both bedrooms and living rooms planning, LayoutGPT attains lower out-of-bound rates than ATISS (bedrooms: 43.26% vs. 49.88%; living rooms: 64.16% vs. 83.02%), which verifies Layout-GPT's spatial reasoning ability in 3D environments. In addition, LayoutGPT has lower FID compared to ATISS (bedrooms: 28.37 vs. 30.02; living rooms: 76.34 vs. 85.40), which indicates that the planned scene has higher quality. Noted here that the living room split contains much more objects on average (11 for living rooms vs. 5 in bedrooms) and is a low-resource split with only 690 training scenes. Therefore, while living rooms are challenging for both methods, LayoutGPT shows more significant improvement over ATISS as supervised methods tend to overfit in early epochs.

Meanwhile, ATISS performs better in terms of KL divergence, which means that the overall furniture distribution predicted by ATISS is closer to the test split. We observe that LayoutGPT tends to avoid furnitures that are extremely rarely seen in each scene (e.g. coffee tables for bedrooms) as these objects appear less frequently in the in-context demonstrations. The limited in-context demonstration size also restricts LayoutGPT to have a universal observation of the furniture distributions.

**Qualitative Results** As shown in Fig. 6, LayoutGPT manages to understand common 3D concepts, such as "the pendant lamp should be suspended from the ceiling" and "nightstands should be placed by the headboard of the bed" (bottom row). When given a floor plan size for both living and dining rooms, LayoutGPT can also generate complicated 3D planning with dining tables and chairs on one side, and a sofa, a coffee table, and a TV stand on the other side (bottom right).

### 5.3 Application Scenarios

**Text-guided Synthesis**: LayoutGPT can follow text captions to arrange furniture in the scene (see Fig. 7). When the captions enumerate a complete list of furniture, LayoutGPT strictly follows the captions to generate the furniture and achieve a KL Div. value close to zero.

**Partial Scene Completion**: Thanks to the autoregressive decoding mechanism, LayoutGPT can complete a scene with partial arrangments such that the additional furniture remains coherent with the existing ones. Through in-context demonstrations, LayoutGPT learns critical (visual) commonsense such as visual symmetric (e.g. nightstands in Fig. 8 (a)), positional relations (e.g. stool at the end of the bed in Fig. 8 (b)), and room functions (e.g. desks and chairs in the dining area in Fig. 8 (d)).

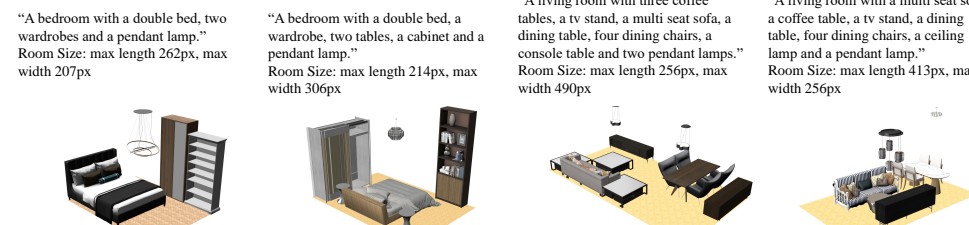

"A bedroom with a double bed, two wardrobes and a pendant lamp." Room Size: max length 262px, max width 207px

"A bedroom with a double bed, a wardrobe, two tables, a cabinet and a pendant lamp." Room Size: max length 214px, max width 306px

"A living room with three coffee tables, a tv stand, a multi seat sofa, a dining table, four dining chairs, a console table and two pendant lamps." Room Size: max length 256px, max width 490px

"A living room with a multi seat sofa, a coffee table, a tv stand, a dining table, four dining chairs, a ceiling lamp and a pendant lamp." Room Size: max length 413px, max width 256px

Figure 7: Generation of 3D scenes based on text captions that enumerate the furniture.

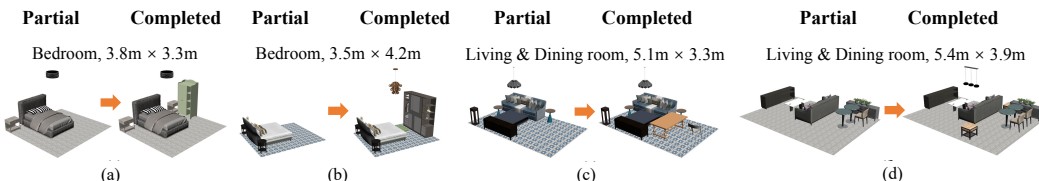

Figure 8: LayoutGPT can successfully complete a partial scene for different rooms. We provide three starting objects for bedrooms and seven objects for living rooms.

## 5.4 Ablation Study

Similar to Sec. 4.4, we study the effect of task instructions, CSS structure, and normalization on indoor scene synthesis (see Table 6). In contrast to our conclusion for 2D planning in Sec. 4.4, comparisons between line 1-4 show that normalization (#4) is the most critical component for suppressing the out-of-bound rate while the CSS structure is also effective. We observe that LLMs occasionally copy attribute values directly from in-context exemplars even though the room sizes are different. Therefore, normalizing all exemplars to the same scale can reduce the out-of-bound rate. CSS style facilitates LLMs to understand the physical meaning behind each attribute value and hence leads to almost the best result when combined with normalization (#7).

Table 6: Ablation studies on LayoutGPT on the bedroom split for 3D indoor scene synthesis.

|   | w/ Instr. | w/ CSS | w/ Norm. | Out of Bound ↓ | KL Div. ↓ | FID ↓ |
|---|---|---|---|---|---|---|
| 1 |   |   |   | 55.32 | 0.1070 | 56.83 |
| 2 | ✓ |   |   | 54.85 | 0.1153 | 58.85 |
| 3 |   | ✓ |   | 51.77 | 0.0776 | 55.62 |
| 4 |   |   | ✓ | 46.57 | 0.1276 | 58.24 |
| 5 | ✓ | ✓ |   | 51.30 | **0.0741** | 57.64 |
| 6 | ✓ |   | ✓ | 46.81 | 0.0913 | 58.61 |
| 7 |   | ✓ | ✓ | 43.74 | 0.0848 | 57.70 |
| 8 | ✓ | ✓ | ✓ | **43.26** | 0.0995 | **56.66** |

## 6 Conclusion

In this work, we address a new direction of generative model collaborations. Specifically, we are interested in how Large Language Models (LLMs) can collaborate with visual generative models. To this end, we propose LayoutGPT, an approach that turns an LLM into a visual planner through in-context learning and CSS style prompts. LayoutGPT can generate plausible visual arrangements in both image space and 3D indoor scenes. LayoutGPT can effectively improve image compositions by generating accurate layouts and achieves comparable performance in indoor scene synthesis compared to supervised methods. Besides, LayoutGPT can improve user efficiency in image generation and serve as an essential part of a unified system for all types of multimodal tasks.

## Acknowledgments

The work was funded by an unrestricted gift from Google and a gift from the Robert N. Noyce Trust to the University of California via the Noyce Initiative. We would like to thank Google and the Robert N. Noyce Trust for their generous sponsorship. The views and conclusions contained in this document are those of the authors and should not be interpreted as representing the sponsors' official policy, expressed or inferred.

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

# A   Implementation Details

In this section, we provide a detailed description of our prompt construction and instantiate instructions examples.

**Task instructions** As is shown in Table 7, the specific task instructions start with verbalized descriptions of the task and are followed by the formal definition of the CSS style. As for the indoor scene synthesis, we additionally provide a list of available furniture and the normalized frequency distribution for fair comparisons with the supervised method. Yet we discover that the provided frequency distribution has little effect on the generation results, based on the trivial change in the KL divergence. In some cases, it is important to make LLMs sample from a defined distribution instead of learning the distribution from in-context exemplars, which we leave for future work.

Table 7: The prepending instructions provided to GPT-3.5/4 during our LayoutGPT's 2D and 3D layout planning process. The instructions listed here are for the setting with CSS structure and with normalization.

| Task | Instruction for GPT-3.5/4 |
|------|---------------------------|
| 2D Layout Planning | Instruction:
Given a sentence prompt that will be used to generate an image, plan the layout of the image. The generated layout should follow the CSS style, where each line starts with the object description and is followed by its absolute position.
Formally, each line should be like "object {width: ?px; height: ?px; left: ?px; top: ?px; }". The image is 64px wide and 64px high. Therefore, all properties of the positions should not exceed 64px, including the addition of left and width and the addition of top and height. |
| 3D Layout Planning | Instruction:
Synthesize the 3D layout of an indoor scene from the bottom-up view. The generated 3D layout should follow the CSS style, where each line starts with the furniture category and is followed by the 3D size, orientation, and absolute position.
Formally, each line should follow the template: FURNITURE {length: ?px: width: ?px; height: ?px; left: ?px; top: ?px; depth: ?px; orientation: ?degrees;} All values are in pixels but the orientation angle is in degrees.

Available furniture: armchair, bookshelf, cabinet, ceiling_lamp, chair, children_cabinet, coffee_table, desk, double_bed, dressing_chair, dressing_table, floor_lamp, kids_bed, nightstand, pendant_lamp, shelf, single_bed, sofa, stool, table, tv_stand, wardrobe
Overall furniture frequencies: (armchair: 0.0045; bookshelf: 0.0076; cabinet: 0.0221; ceiling_lamp: 0.062; chair: 0.024; children_cabinet: 0.0075; coffee_table: 0.0013; desk: 0.0172; double_bed: 0.1682; dressing_chair: 0.0063; dressing_table: 0.0213; floor_lamp: 0.0093; kids_bed: 0.0079; nightstand: 0.2648; pendant_lamp: 0.1258; shelf: 0.0086; single_bed: 0.0211; sofa: 0.0018; stool: 0.012; table: 0.0201; tv_stand: 0.0308; wardrobe: 0.1557) |

**Base LLMs** We use four variants of GPT models, (1) Codex [5] (`code-davinci-002`), an LLM that is fine-tuned with large-scale code datasets and can translate natural language into functioning code snippets; (2) GPT-3.5 [36] (`text-davinci-003`), which is trained to generate text or code from human instructions; (3) GPT-3.5-chat (`gpt-3.5-turbo`) and (4) GPT-4 [35] (`gpt-4`), which are both optimized for conversational tasks. For the last two models, we first feed the in-context exemplars as multiple turns of dialogues between the user and the model to fit into the API design. However, we generally observe that GPT-3.5-chat and GPT-4 are not as strong as GPT-3.5 in learning from the in-context demonstrations, especially when the dialogue format follows a certain structure instead of free-form descriptions.

**Hyperparameters** For all LLMs, we fix the sampling temperature to 0.7 and apply no penalty to the next token prediction. For image layouts evaluation in Table 2, we fix the number of exemplars to 16 for numerical reasoning, and 8 for spatial reasoning, based on the best results of a preliminary experiment. However, we do not observe significant gaps in evaluation results when using different amounts of exemplars (see Sec. B.4). For each prompt, we generate five different layouts/images using baselines or LayoutGPT and thus result in 3810 images for numerical reasoning and 1415 images for spatial reasoning in all reported evaluation results. As for indoor scene synthesis, we fix the number of exemplars to 8 for bedrooms and 4 for living rooms to reach the maximum allowed input tokens. We set the maximum output token as 512 for bedrooms and 1024 for living rooms as bedrooms have ∼5 objects per room while living rooms have ∼11 objects per room. We generate one layout for each rectangular floor plan for evaluation.

# B    LayoutGPT for 2D Layout Planning

## B.1    NSR-1K Benchmark Construction

We rely on the MSCOCO annotations to create NSR-1K with ground-truth layout annotations. Note that each image in COCO is paired with a set of captions and a set of bounding box annotations.

**Numerical Reasoning**    We primarily focus on the competence of T2I models to count accurately, i.e., generate the correct number of objects as indicated in the input text prompt. The prompts for this evaluation encompass object counts ranging from 1 to 5. To design the template-based T2I prompts, we initially sample possible object combinations within an image based on the bounding box annotations. We only use the bounding box annotation of an image when there are at most two types of objects within the image. As a result, the template-based prompts consist of three distinct types: (1) *Single Category*, wherein the prompt references only one category of objects in varying numbers; (2) *Two Categories*, wherein the prompt references two categories of distinct objects in varying numbers; and (3) *Comparison*, wherein the prompt references two categories of distinct objects but specifies the number of only one type of object, while the number of the other type is indicated indirectly through comparison terms including "fewer than", "equal number of", and "more than". As for natural prompts, we select COCO captions containing one of the numerical keywords from "one" to "five" and filter out those with bounding box categories that are not mentioned to avoid hallucination.

**Spatial Reasoning**    We challenge LLMs with prompts that describe the positional relations of two or more objects. Our spatial reasoning prompts consist of template-based prompts and natural prompts from COCO. To construct template-based prompts, we first extract images with only two ground-truth bounding boxes that belong to two different categories. Following the definitions from PaintSkill [7], we ensure the spatial relation of the two boxes belong to (left, right, above, below). Specifically, given two objects $A, B$, their bounding box centers $(x_A, y_A), (x_B, y_B)$ and the Euclidean distance $d$ between two centers, we define their spatial relation $\text{Rel}(A, B)$ as:

$$\text{Rel}(A, B) = \begin{cases} B \text{ above } A & \text{if } \frac{y_B - y_A}{d} \geqslant sin(\pi/4) \\ B \text{ below } A & \text{if } \frac{y_B - y_A}{d} \leqslant sin(-\pi/4) \\ B \text{ on the left of } A & \text{if } \frac{x_B - x_A}{d} < cos(3\pi/4) \\ B \text{ on the right of } A & \text{if } \frac{x_B - x_A}{d} > cos(\pi/4) \end{cases} \tag{1}$$

The definition basically dissects a circle centered at $A$ equally into four sectors that each represent a spatial relation. While the definition may not stand for all camera viewpoints, it allows us to mainly focus on the **front view** of the scene. Then, we utilize the category labels and the pre-defined relations to form a prompt, as is shown in Table 1. As for the natural COCO prompts, we select prompts that contain one of the key phrases (the left/right of, on top of, under/below) and ensure that the bounding box annotations align with our definition.

## B.2    Evaluation Metrics

We denote the set of $n$ object categories in the ground truth annotation as $\mathcal{C}_{GT} = c_1, c_2, \ldots, c_n$, where $x_{c_1}, x_{c_1}, \ldots, x_{c_n}$ represent the number of objects for each category. Additionally, we denote the set of $m$ object categories mentioned in GPT-3.5/4's layout prediction as $\mathcal{C}_{pred} = c'_1, c'_2, \ldots, c'_m$, where $x'_{c'_1}, x'_{c'_2}, \ldots, x'_{c'_m}$ represent the number of objects for each category accordingly. If a category $c_i$ is not mentioned in $\mathcal{C}_{pred}$, then $x'_{c_i}$ is assigned a value of 0, and vice versa.

| Categories | $c_i$ | cat | bed | pillow |
|---|---|---|---|---|
| **Ground Truth** | $x_{c_i}$ | 2 | 1 | 2 |
| **Prediction** | $x'_{c_i}$ | 1 | 0 | 3 |

$$precicion = \frac{\sum min(x_{c_i}, x'_{c_i})}{\sum x'_{c_i}} = \frac{1 + 0 + 2}{1 + 0 + 3} = 75\%$$

$$recall = \frac{\sum min(x_{c_i}, x'_{c_i})}{\sum x_{c_i}} = \frac{1 + 0 + 2}{2 + 1 + 2} = 60\%$$

Figure 9: An closeup example of how we compute the layout automatic evaluation metrics for numerical reasoning.

Table 8: Closeup of various in-context example formats with ablated CSS structure and normalization for 2D layout planning.

| CSS Structure | Normalization | In-context Example Format Demo |
|:---:|:---:|---|
| | | Prompt: a teddy bear to the right of a book
Layout:
teddy bear: 0.50, 0.71, 0.50, 0.15
book: 0.50, 0.61, 0.00, 0.26 |
| ✓ | | Prompt: a teddy bear to the right of a book
Layout:
teddy bear {width: 0.50; height: 0.71; left: 0.50; top: 0.15; }
book {width: 0.50; height: 0.61; left: 0.00; top: 0.26; } |
| | ✓ | Prompt: a teddy bear to the right of a book
Layout:
teddy bear: 32, 45, 31, 9
book: 31, 38, 0, 16 |
| ✓ | ✓ | Prompt: a teddy bear to the right of a book
Layout:
teddy bear {width: 32px; height: 45px; left: 31px; top: 9px; }
book {width: 31px; height: 38px; left: 0px; top: 16px; } |

The numerical reasoning ability of GPT-3.5/4 on layout planning is assessed using the following metrics: (1) *precision*: calculated as $\frac{\sum_{k=1}^{n} \min(x_{c_k}, x'_{c_k})}{\sum_{k=1}^{m} x'_{c'_k}}$, is an indication of the percentage of predicted objects that exist in the groundtruth; (2) *recall*: calculated as $\frac{\sum_{k=1}^{n} \min(x_{c_k}, x'_{c_k})}{\sum_{k=1}^{n} x_{c_k}}$, indicates the percentage of ground-truth objects that are covered in the prediction; (3) *accuracy*: In the "comparison" subtask, an accuracy score of 1 is achieved when the predicted relation, whether it is an inequality or equality, between the two objects is accurately determined. For all other numerical subtasks, accuracy equals to 1 if the predicted categories and object numbers precisely match the ground truth. In other cases, the accuracy is 0. Fig. 9 shows an example of how we compute the *precision* and *recall*. The *accuracy* for this single example is 0 since the predicted object distribution does not match the ground truth in every category.

For spatial reasoning, we evaluate spatial accuracy based on the LLM-generated layouts and GLIP-based layouts. We adopt [26] finetuned on COCO to detect involved objects from the generated images and obtain the bounding boxes. For both types of layouts, we categorize the spatial relation based on the above definition and compute the percentage of predicted layouts with the correct spatial relation. For all evaluation benchmarks, we measure the CLIP similarity, which is the cosine similarity between the generated image feature and the corresponding prompt feature.

### B.3   GPT-3.5/4 Prompting

In Sec. 4.4, we investigate the impact of three components in the structured prompts: (1) *Instruction*, which examines whether detailed instructions explaining the task setup and the format of the supporting examples are included in the prompt. (2) *Structure*, which evaluates the impact of different formatting settings on the presentation of the bounding box aspects of height, width, top, and left. The "w/ CSS" setting formats the aspects in CSS, while the "w/o CSS" setting presents the four aspects in a sequence separated by a comma. (3) *Normalization*, which investigates the effects of rescaling the bounding box aspects to a specified canvas size and presenting them as integers in pixels in the "w/ Norm." setting, while the "w/o Norm." setting presents the aspects as relative scales to the canvas size in floats that range from (0, 1).

Table 7 shows the detailed prepending instructions LayoutGPT provided to GPT-3.5/4 models during 2D layout planning. Table 8 compares the formats of supporting examples with ablated structures and normalization settings.

For experiments in Sec. 4.3, to adapt to the nature of the Panoptic task, we add the following additional instruction when prompting LayoutGPT: *"The objects layout might be dense, and objects may overlap with each other. Some objects might not have been mentioned in the prompt, but are very likely to appear in the described scenario."* To generate counterfactual prompts for text-to-image generation, the following text prompt is provided to GPT-4: *"Please provide a few counterfactual prompts that*

Table 9: The automatic metric scores of LayoutGPT (GPT-3.5) with different in-context sample selection approaches. All values are in percentage (%).

| # | Exemplar Selection | # In-Context Exemplars | Numerical Reasoning | | | | Spatial Reasoning | |
|---|---|---|---|---|---|---|---|---|
| | | | Precision↑ | Recall↑ | Layout Accuracy↑ | GLIP Accuracy↑ | Layout Accuracy↑ | GLIP Accuracy↑ |
| 1 | Fixed Random | 16 | 64.83 | 92.71 | 87.66 | 47.10 | 80.14 | 47.07 |
| 2 | Retrieval | 4 | 88.93 | 95.02 | 76.17 | 50.20 | 85.30 | 51.66 |
| 3 | | 8 | 93.32 | 95.63 | 82.68 | 50.58 | 82.54 | 52.86 |
| 4 | | 16 | 94.81 | 96.49 | 86.33 | 51.25 | 82.40 | 51.09 |

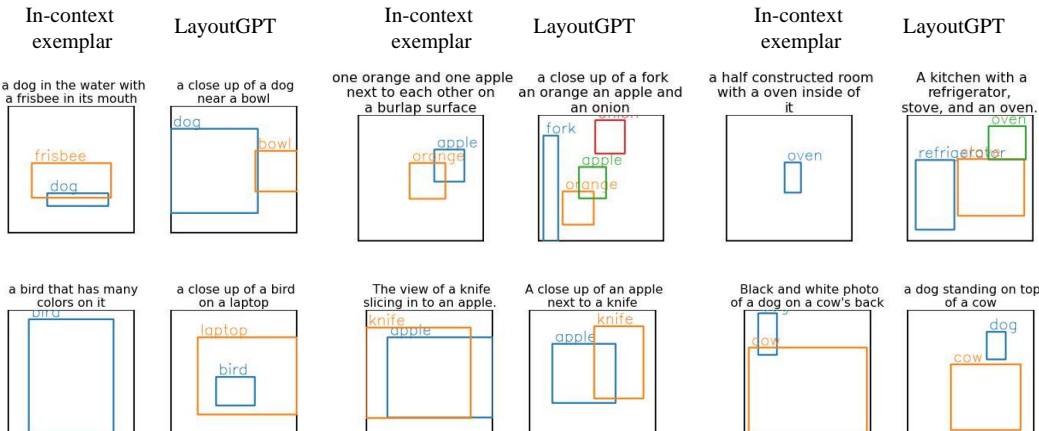

Figure 10: Comparison between the most similar in-context exemplar and the generation results of LayoutGPT.

*depict rarely seen the spatial relationship between the 80 MSCOCO object categories. An example would be "a monkey riding on top of a bird""*.

## B.4 Additional Experiments

**Random In-Context Exemplars**  Empirically, selecting in-context exemplars can be critical for the overall performance of LLMs. Apart from our retrieval-augmented method in Sec. 3, we also experiment with a **fixed random** set of in-context exemplars. Specifically, we randomly sample $k$ examples from the training (support) set $D$ to form a fixed set of in-context demonstrations for all test conditions $\mathcal{C}_j$. Therefore, the fixed random setting results in in-context exemplars that are unrelated to the test condition $\mathcal{C}_j$. The minor gap between lines 1&5 in Table 9 verifies that LayoutGPT is not directly copying from the in-context exemplars in most cases. Fig. 10 further justifies the argument with layout visualization of the most similar in-context exemplars and the LayoutGPT outputs.

**Number of In-Context Exemplars**  We take a closer look at the effects of the number of in-context exemplars in the prompt as shown in Table 9. For counting, we observe that the number of exemplars is positively correlated with the counting accuracy. We conjecture that LLMs learn to make more accurate predictions for challenging prompts (e.g., comparison) by learning from more few-shot exemplars. As the layout accuracy also accounts for results where CSS parsing fails, we observe that the LLMs generate more consistent CSS-style code by learning from more examples. However, we cannot observe a similar trend in spatial reasoning prompts. We conjecture that LLMs only require as few as four demonstrations to learn the differences between the four types of spatial relations. The small optimal number of in-context exemplars implies that LLMs already have 2D spatial knowledge and can map textual descriptions to corresponding coordinate values. Yet it is important to find a proper representation to elicit such knowledge from LLMs as implied in Sec. 4.4.

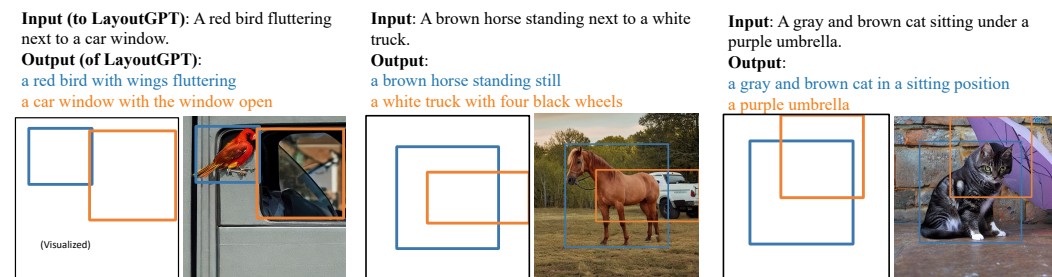

Figure 11: Attribute binding examples of LayoutGPT and generated images using ReCo.

Table 10: The layout performance on each numerical reasoning subtask. Results reported on LayoutGPT (GPT-4).

| Prompt Source | Subtask | Precision | Recall | Accuracy |
|---|---|---|---|---|
| | Single Category | 85.96 | 85.96 | 85.96 |
| Template | Two Categories | 85.14 | 85.04 | 66.60 |
| | Comparison | - | - | 77.80 |
| Natural Prompts from MSCOCO | | 72.08 | 87.1 | 82.79 |
| - | Total | 78.36 | 86.29 | 78.43 |

**Performance on Numerical Subtasks**  Table 10 presents the performance of layout generation in various numerical reasoning subtasks. Regarding template-based prompts, the LayoutGPT demonstrates superior performance in the "Single Category" numerical reasoning task, exhibiting precision, recall, and accuracy values around 86%. However, when it comes to the "Two Category" numerical reasoning task, while precision and recall experience minimal changes, the accuracy drops to 66%. For the "Comparison" subtask, the accuracy hovers around 78%. These outcomes indicate that LayoutGPT encounters greater challenges when confronted with multi-class planning scenarios, whether the number of objects is explicitly provided or indirectly implied through comparative clauses.

For natural prompts extracted from MSCOCO, a noteworthy observation is the high recall accompanied by relatively lower precision. This discrepancy arises due to the ground truth bounding box annotations encompassing only 80 object classes, whereas the natural prompts may mention objects beyond the annotated classes. Consequently, our LayoutGPT may predict object layouts corresponding to classes not present in the ground truth, which, despite lowering precision, aligns with the desired behavior.

**Performance on Size Comparison Reasoning Task**  We evaluate LayoutGPT's reasoning ability regarding object size. We use the standard HRS benchmark [2] which is designed for benchmarking compositional text-to-image models. HRS prompts for size reasoning contain comparison terms between randomly sampled common objects. The size relations described in HRS size prompts are often counterfactual and rarely seen (e.g., *"a person which is smaller than a chair and larger than horse"*, *"a car which is smaller than a banana and chair and bigger than airplane"*.). LayoutGPT achieves an accuracy of 98.0% / 93.1% / 92.1% when the prompt involves size comparison between 2/3/4 objects. Meanwhile, the best size reasoning performance of nine text-to-image models reported by the HRS benchmark is only 31.1% / 0.2% / 0%. The results further verify that LayoutGPT acquires decent reasoning ability on rare scenarios / counterfactual prompts.

## B.5  Failure Cases

Fig. 12 shows typical failure cases in numerical and spatial relations. As previously discussed, we observe in Table 10 that numerical prompts that involves two type of objects ("Two Categories" and "Comparison") are more challenging to LayoutGPT and the image generation model. In these subtasks, LayoutGPT tends to predict much smaller bounding boxes to fit all objects within the limited image space. The small boxes further challenge GLIGEN to fit the object within the limited region, as shown in Fig. 12 (right).

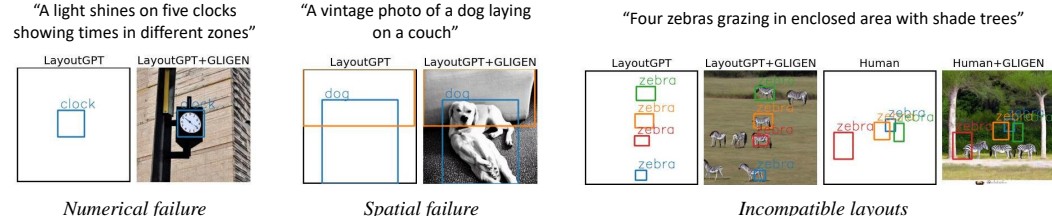

*Numerical failure*     *Spatial failure*     *Incompatible layouts*

Figure 12: Typical failure cases of LayoutGPT and the generation results using GLIGEN.

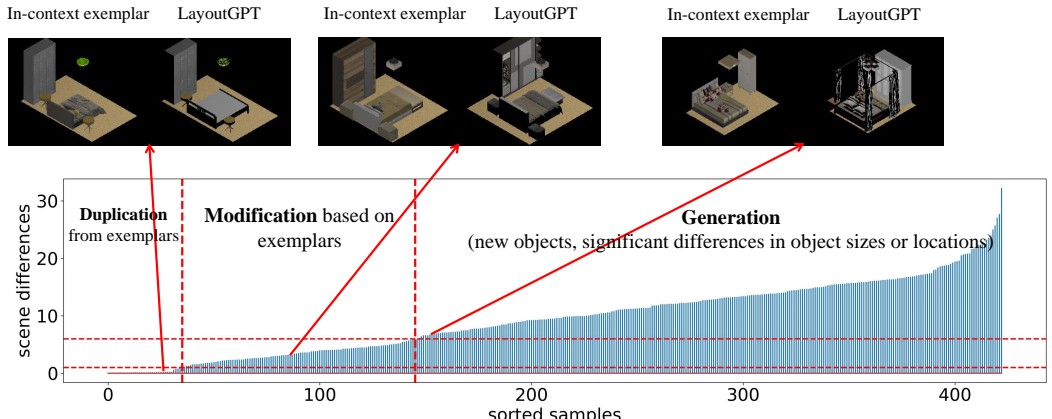

Figure 13: Sorted scene differences between LayoutGPT generated scenes and the most similar in-context exemplars of 423 testing bedroom samples. We partition the distribution into three segments representing different behaviors of LayoutGPT. Duplication: The generated scene is a duplication of the exemplar. Modification: LayoutGPT slightly modifies one exemplar as the generated layout. Generation: LayoutGPT generates novel scenes that are highly different from the exemplars.

## C  LayoutGPT for 3D Scene Synthesis

Due to the limitation in datasets, the conditions are room type and room size instead of text descriptions. While ATISS [38] utilizes the floor plan image as the input condition, LLMs are not compatible with image inputs. Therefore, we convert the floor plan image into the specification of the room size. Therefore, the input conditions are similar to "*Room Type: Bedroom, Room Size: max length 256px, max width 256px*".

### C.1  Exemplar Selection

Similar to Sec. B.4, we investigate the effect of using a random set of in-context exemplars for indoor scene synthesis. When we apply 8 random bedroom layouts from the training set as in-context exemplars, the out-of-bound rate increases from 43.26% in Table 5 to 85.58%. The significant differences suggest that LayoutGPT heavily relies on rooms with similar floor plans to maintain objects within the boundary. Yet we verify that the generated layouts from LayoutGPT are not duplicates of the in-context exemplars in most cases.

We first define a training scene layout as a set of objects $S^t = \{\mathbf{o}_1^t, \ldots, \mathbf{o}_m^t\}$, and a generated scene layout as $S^g = \{\mathbf{o}_1^g, \ldots, \mathbf{o}_n^g\}$. Note that $\mathbf{o}_j$ consists of category $\mathbf{c}_j$, location $\mathbf{t}_j \in \mathbb{R}^3$, size $\mathbf{s}_j \in \mathbb{R}^3$, and orientation $\mathbf{r}_j \in \mathbb{R}$, i.e. $\mathbf{o_j} = (\mathbf{c}_j, \mathbf{t}_j, \mathbf{s}_j, \mathbf{r}_j)$ We define the scene difference $D(\cdot|\cdot)$ between $S^t$ and $S^t$ as

$$D(S^t|S^g) = \sum_{i=1}^{n} \min_{j, \mathbf{c}_j^t = \mathbf{c}_i^g} (\|\mathbf{t}_j^t - \mathbf{t}_i^g\|_1 + \|\mathbf{s}_j^t - \mathbf{s}_i^g\|_1). \qquad (2)$$

We set $\mathbf{t}_j^t, \mathbf{s}_j^t$ to $\mathbf{0}$ if $S^t$ does not have a single object that belongs to the same category as $\mathbf{c}_i^g$. For each testing sample of the bedroom, we compute the scene differences between the generated layout

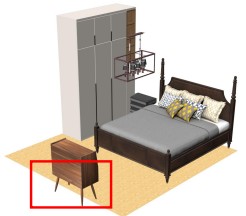 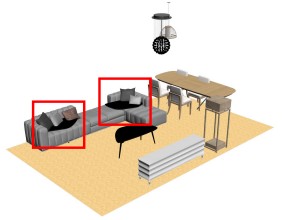 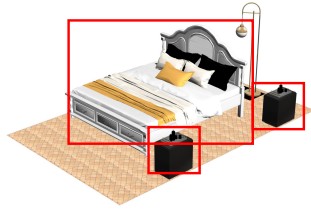

| Out-of-Bound Furniture | Overlapped Objects | Inharmonious placement |

Figure 14: Typical failure cases of LayoutGPT.

and all eight in-context exemplars and use the minimum value as the final scene difference. Note that all parameters used for computation are in "meters" instead of "pixels".

We plot the scene differences of all 423 testing samples in Fig. 13. We empirically discover that a scene difference below 1.0 means $S^g$ is highly similar to $S^t$, which we conclude as **duplication** from in-context exemplars. A scene difference below 6.0 shows moderate differences in object sizes or locations between two scenes, representing a **modification** based on $S^t$ to generate $S^g$. Finally, a scene difference larger than 6.0 represents new objects or significant differences in object sizes or locations between the exemplar and the generated layouts, i.e. true **generation**. Fig. 13 shows that 34/111/278 scenes belong to duplication/modification/generation. Among each category, 30/67/143 scenes have no out-of-bound furniture. Therefore, LayoutGPT is performing generation instead of duplicating in-context exemplars in most cases.

## C.2 Failure Cases

While LayoutGPT achieves comparable results as ATISS, LayoutGPT cannot avoid typical failure cases as shown in Fig. 14, such as out-of-bound furniture and overlapped objects. Fig. 14 (right) shows an incorrect placement of nightstands on the same side of the bed while they are commonly placed on each side of the bed headboard. Future work could focus on more sophisticated in-context learning or fine-tuning methods to improve the LLMs' understanding of 3D concepts.

# D LayoutGPT for 2D Keypoint Planning

In addition to its application in 2D and 3D layout planning, we investigate the feasibility of leveraging LayoutGPT for 2D keypoint planning to facilitate text-conditioned image generation. In this approach, we utilize LayoutGPT to predict keypoint distributions based on a given text prompt, and subsequently employ GLIGEN [27] for keypoint-to-image generation. The keypoint format used aligns with the specifications outlined in MSCOCO2017 [29], focusing on 17 keypoints that correspond to the human skeleton. Similar to our methodology for selecting supporting examples in the context of 2D layout planning (Section B), we retrieve the $k$-most similar examples from the training set of MSCOCO2017 and utilize these examples to provide keypoint distributions as input to GPT-3.5/4. Table 11 presents an illustrative example of the input format employed for keypoint planning with GPT-3.5.

Fig. 15 presents several illustrative examples that compare the images generated by conditioning on keypoints planned by our LayoutGPT with those generated by end-to-end models such as StableDiffusion-v2.1 [41] and Attend-and-Excite [4]. In this preliminary demonstration, we observe that LayoutGPT exhibits promising potential in offering inherent control over specific movements or actions through keypoint planning.

Nevertheless, it is worth noting that keypoints planning presents considerably greater challenges compared to bounding box layout planning, attributable to several evident factors. Firstly, keypoints planning necessitates the prediction of the positions of 17 nodes, which is significantly more complex than the 2D layout planning involving four aspects or the 3D layout planning encompassing seven aspects. Secondly, the distribution of keypoints encompasses a much larger array of spatial relations due to the numerous possible body movements. In contrast, previous 2D layout planning tasks only involve four types of spatial relations. These inherent complexities render keypoint planning heavily reliant on in-context demonstrations. However, the limited availability of annotations pertaining to

Table 11: The prompting input provided to GPT-3.5 for LayoutGPT keypoint planning.

Instruction:
Given a sentence prompt that will be used to generate an image, plan skeleton keypoints layout of the mentioned objects. The skeleton keypoints include the following 17 nodes: nose, left_eye, right_eye, left_ear, right_ear, left_shoulder, right_shoulder, left_elbow, right_elbow, left_wrist, right_wrist, left_hip, right_hip, left_knee, right_knee, left_ankle, right_ankle. The generated keypoints layout should follow the CSS style, where each line starts with the keypoint node name and is followed by its absolute position.
Formally, each line should be like "node_name {left: ?px; top: ?px; }". Please follow this format strictly. Do not display in other variation of formats. Notice that some keypoint nodes may not be visible on the canvas. In such cases, simply put "node_name {left: 0px; top: 0px; }" for the invisible nodes. The image is 64px wide and 64px high. Therefore, all properties of the positions should not exceed 64px.

Prompt: a man on a surfboard in a river near a couple of trees and branches
Keypoints:
person#1:
nose {left: 36px; top: 33px; }
left_eye {left: 36px; top: 33px; }
right_eye {left: 36px; top: 33px; }
left_ear {left: 37px; top: 33px; }
right_ear {left: 0px; top: 0px; }
left_shoulder {left: 38px; top: 34px; }
right_shoulder {left: 36px; top: 35px; }
left_elbow {left: 35px; top: 34px; }
right_elbow {left: 35px; top: 38px; }
left_wrist {left: 33px; top: 32px; }
right_wrist {left: 33px; top: 39px; }
left_hip {left: 39px; top: 39px; }
right_hip {left: 37px; top: 40px; }
left_knee {left: 38px; top: 44px; }
right_knee {left: 37px; top: 44px; }
left_ankle {left: 39px; top: 49px; }
right_ankle {left: 37px; top: 48px; }

[MORE SUPPORTING EXAMPLES]

Prompt: a man leaning on a surfboard in the water riding a wave
Keypoints:

body movements in the MSCOCO dataset further exacerbates the challenges associated with reliable keypoint planning. Therefore, we leave the exploration of this potential direction to future research endeavors.

# E   Ethical Statement

In addition to the layouts predicted by GPT-3.5/4, we also incorporate human-planned layouts as a natural baseline for comparative analysis. To facilitate this, we provide annotators with an interface featuring a blank square space where they can draw bounding boxes. Alongside the input text prompt, we also present the noun words or phrases from the prompt to human annotators, instructing them to draw a bounding box for each corresponding element. We intentionally refrain from imposing additional constraints, enabling annotators to freely exercise their imagination and create layouts based on their understanding of reasonable object arrangements. To compensate annotators for their efforts, we offer a payment rate of $0.2 US dollars per Human Intelligence Task (HIT). The average completion time of approximately 30 seconds per HIT, which corresponds to an average hourly payment rate of $24.

# F   Limitations

The current work has several limitations that provide opportunities for future research. Firstly, while this work focuses on 2D and 3D bounding box layouts and makes a preliminary attempt at keypoints, there exist various other methods for providing additional spatial knowledge in image/scene

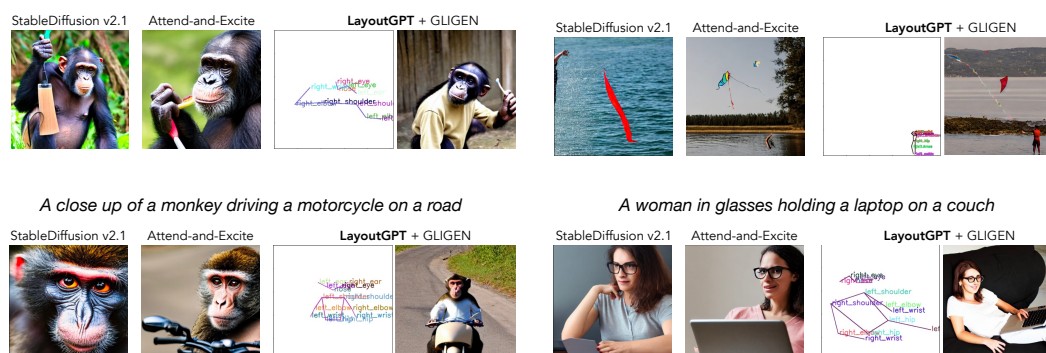

Figure 15: Plausible examples of LayoutGPT(GPT-4) planning keypoints distributions before conducting text-conditioned image generation.

generation, such as segmentation masks and depth maps. Future work could explore integrating LLMs with these alternative visual control mechanisms to broaden the scope of visual planning capabilities. Secondly, the current work primarily addresses visual generation tasks and lacks a unified framework for handling other visual tasks like classification or understanding. Extending the proposed framework to encompass a wider range of visual tasks would provide a more comprehensive and versatile solution. Thirdly, this work is a downstream application that attempts to distill knowledge from LLMs' extensive knowledge bases. Future research could explore more fundamental approaches that directly enhance the visual planning abilities of various visual generation models. By developing specialized models that are explicitly designed for visual planning, it may be possible to achieve more refined and dedicated visual generation outcomes. Overall, while the current work demonstrates the potential of using LLMs for visual planning, there are avenues for future research to address the aforementioned limitations and further advance the field of visual generation and planning.

## G   Broader Impact

The utilization of LLMs for conducting visual planning in compositional 2D or 3D generation has significant broader impacts. Firstly, LLMs alleviate the burden on human designers by simplifying the complex design process. This not only enhances productivity but also facilitates scalability, as LLMs can efficiently handle large-scale planning tasks. Secondly, LLMs exhibit remarkable capabilities in achieving fine-grained visual control. By conditioning on textual inputs, LLMs can easily generate precise and detailed instructions for the desired visual layout, allowing for precise composition and arrangement of elements. Moreover, LLMs bring a wealth of commonsense knowledge into the planning process. With access to vast amounts of information, LLMs can incorporate this knowledge to ensure more accurate and contextually coherent visual planning. This integration of commonsense knowledge enhances the fidelity of attribute annotations and contributes to more reliable and realistic visual generation outcomes.

It is worth noting that this work represents an initial foray into the realm of visual planning using LLMs, indicating the potential for further advancements and applications in this area. As research in this field progresses, we can anticipate the development of more sophisticated and specialized visual planning techniques, expanding the scope of LLMs' contribution to diverse domains, such as architecture, virtual reality, and computer-aided design.

## H   Additional Qualitative Examples

We present additional visual showcases to demonstrate the capabilities of LayoutGPT in different contexts. Fig. 16 showcases examples related to 2D numerical reasoning, Fig. 17 illustrates examples of 2D spatial reasoning, and Fig. 18 displays examples of 3D scene synthesis. These showcases offer further insights into the effectiveness and versatility of our approach across various domains.

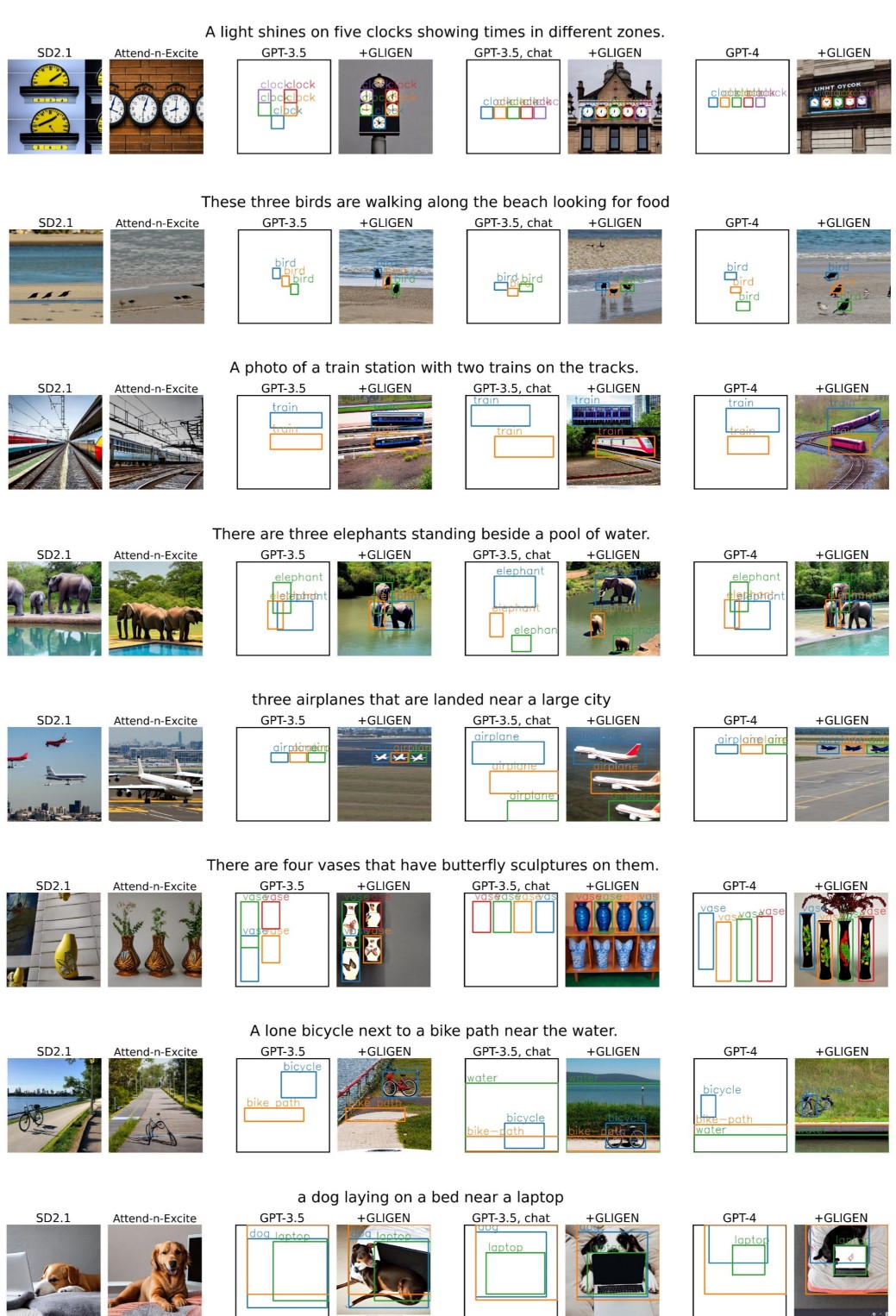

Figure 16: Qualitative examples of variants of LayoutGPT on numerical reasoning prompts.

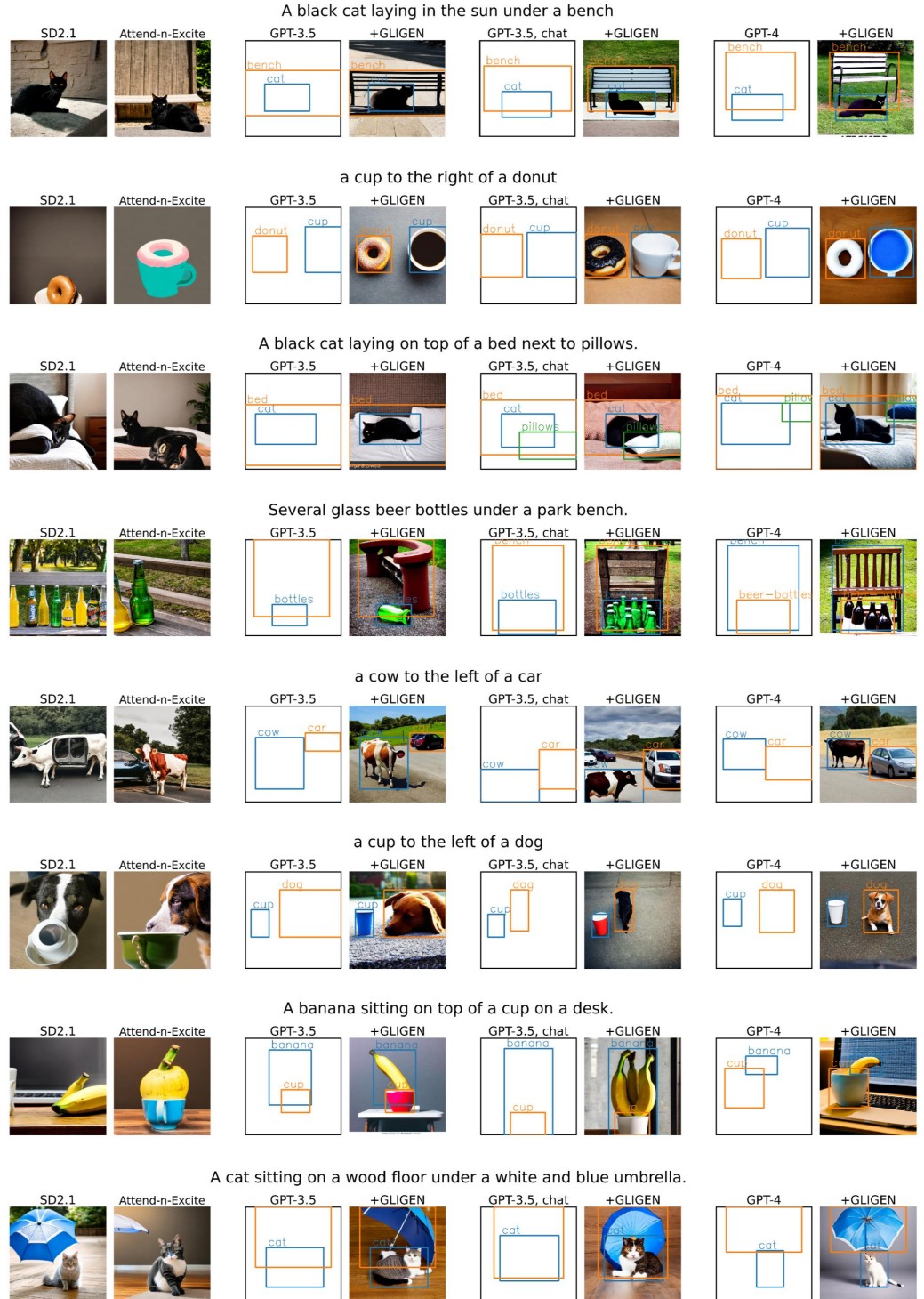

Figure 17: Qualitative examples of variants of LayoutGPT on spatial reasoning prompts.

GPT-3.5             GPT-3.5-chat             GPT-4

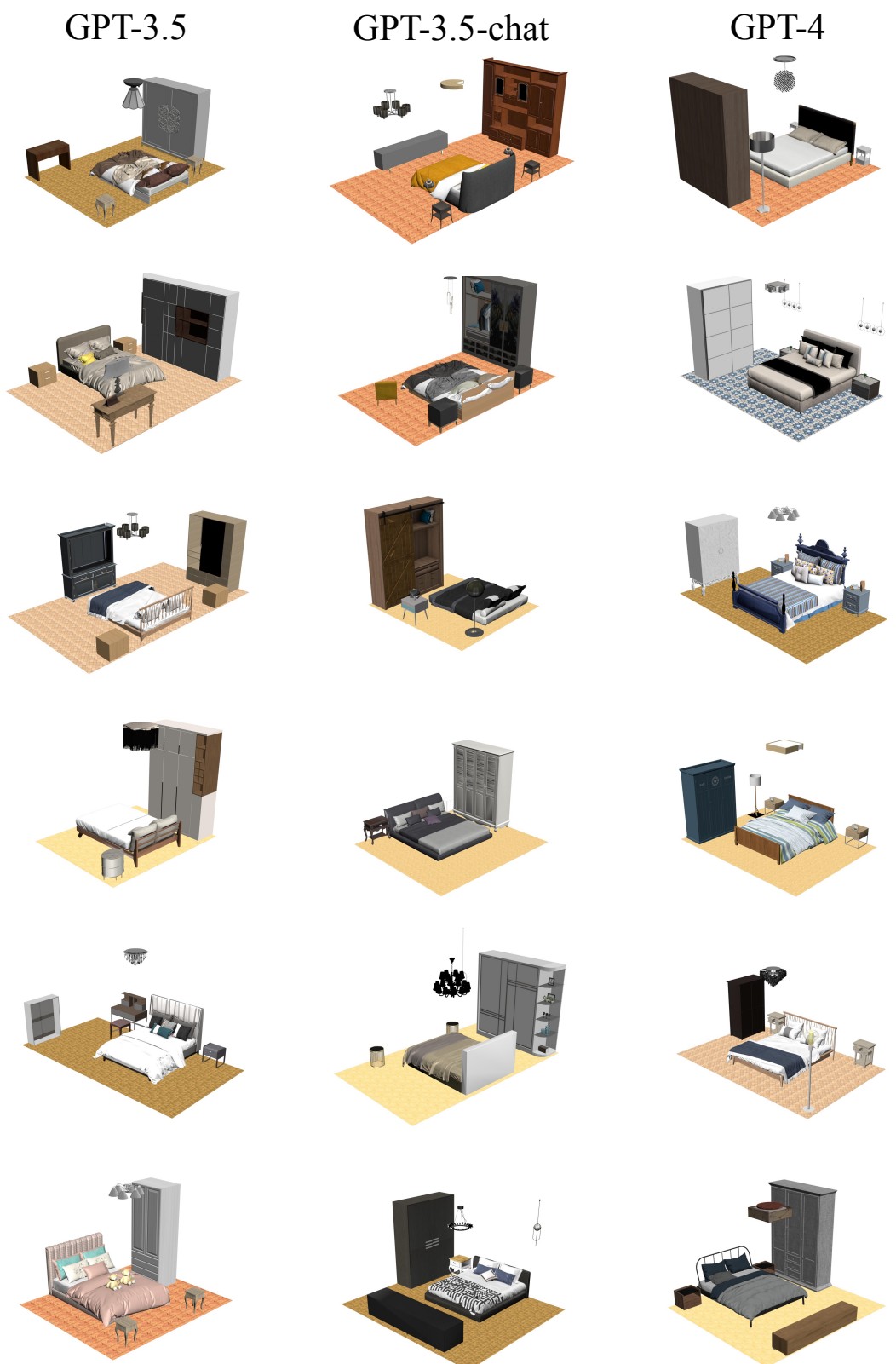

Figure 18: Additional qualitative examples of variants of LayoutGPT in bedroom scene synthesis.

