# OpenReview forum: "LayoutGPT: Compositional Visual Planning and Generation with Large Language Models"
_NeurIPS.cc/2023/Conference — NeurIPS 2023 poster_

### Official Review · Reviewer_uKyZ · 2023-06-25

**Soundness:** 3 good
**Presentation:** 2 fair
**Contribution:** 2 fair
**Rating:** 5
**Confidence:** 3

**Summary:**

The paper highlights the challenges faced by existing models in generating objects with specified counts, positions, attributes, and sizes, and emphasizes the need for compositional skills that can effectively arrange components coherently, accurately reflecting object specifications and interactions.
The authors propose LayoutGPT which aims to improve the utility of visual planning skills of Large Language Models (LLMs) by in-context learning.
The method shows promising results in generating plausible layouts in multiple domains including 2D images and 3D indoor scenes.
Experiments show that LayoutGPT outperforms text-to-image models and achieves comparable performance to human users in designing visual layouts for numerical and spatial reasonings.
In addition, LayoutGPT also shows comparable performance to supervised methods in 3D indoor scene synthesis.
The paper also proposes a new benchmark called NSR-1K for evaluating generations in terms of specified counts and spatial locations.


**Strengths:**

1. The authors propose a novel solution for handling the issue that the current visual; generation models lack visual arrangments ability. They adopt LLMs as visual planners to generate layout information of objects in the target 2D images or 3D scenes, with the help of in-context visual demonstrations in style sheet language.
2. LayoutGPT generates reasonable layouts in multiple domains, including 2D images and 3D indoor scenes.
3. They build a challenging benchmark that characterizes counting and positional relations for text-to-image generations.
4. Experiments show the effectiveness of the proposed method.
5. The paper presents promising results and provides insights into the application of LLMs for visual planning and generation, both in terms of methodology and empirical findings.

**Weaknesses:**


1. The writing of motivation is not clear due to a lot of inexplicable expressions.

    - The claim that existing visual generative models are not equipped with various reasoning skills that exist in LLMs is dubious, given the disparate nature of reasoning skills required for visual generative models compared to LLMs. Good reasoning skills are not equal to good generative effects. Therefore, the T2I models that fail to generate objects with specific counts, positions, and attributes do not indicate lacking reasoning skills.

    - In the sentence, 'But unlike LLMs, ... discrete categories ...', do the authors mean LLMs are already capable of generating layout? if yes, there should have some citations, and there is no need to further introduce the LLMs as a centralized model because there is no logical correlation with the previous text. In addition, why did the authors strengthen the discrete categories, could the LLMs generate continuous categories?

    - Unclear description of the drawbacks of the existing LLMs-centered systems.

    - The motivations behind constructing layouts with structured programs are not sufficiently persuasive, considering that LLMs also are trained on the plain text, and alternative methods exist for representing image layouts. In fact, the strict imposition of a structured format may pose challenges for LLMs.

    - To my knowledge, visual inputs generally refer to images or videos, and the LLMs are only able to handle textual inputs. Moreover, both tasks in this work involve textual information as input, and there is no experimental evidence supporting the claim that LLMs have the potential in handling complex visual inputs.


    - The meaning of "addressing the inherent multimodal reasoning skills of LLMs" requires clarification. To my understanding, LLMs already possess visual planning capabilities, and the authors' objective is to effectively leverage these skills rather than fundamentally modify or address them.


2. Inconsistent notations.

    - $o_j$ vs $\mathbf{o}_j$ in the section 3.1.

    - $o_j$ represents the layout of an object, so the sequence should be (c_j, x_j, y_j, w_j, h_j).



3. The utilization of CSS as a structured format for representing layouts is a bit farfetched and unnecessary. Firstly, LLMs are capable of understanding the meaning of each element in a plain sequence (o) if detailed task instruction is provided. Secondly, structured representation is not solely limited to the CSS format; alternative formats such as, 'layout: c_1(x_1, y_1, w_1, h_1), c_2(x_2, y_2, w_2, h_2), ...'.


4. The reasons behind the lower performance observed when GLIGEN generates images based on ground truth (GT) layouts during image evaluation, as well as when GLIGEN synthesizes images using layouts provided by humans, remain unexplained in Table 2.


5. It is important to note that LayoutGPT is specifically designed for generating layouts, encompassing object categories and corresponding bounding boxes. Therefore, it is unreasonable to conclude that LayoutGPT can accurately perform attribute binding. Correct attribute binding is only achieved when the attributes of objects in the generated images align with the textual descriptions. However, wrong attribute bindings between the generated images and sentences are apparent, such as the closed car window in the top row of Figure 4 and the absence of a handle on the basket in the bottom row of Figure 4.




**Questions:**

1. The authors mention that the in-context examples are provided in reverse order based on similarity. Does the order have an impact on the effectiveness of the model？
2. How are the colored sentences in Figure 4 generated?
3. For 3D layout planning, how do the furniture frequencies in the instruction impact the performance?
4. Why does the CSS structure have more impact on task performance compared to instruction?



**Limitations:**

I do not see any limitations to this work.

---

> ### Author Rebuttal · Authors · 2023-08-10
>
> We thank the reviewer for pointing out ambiguities in the manuscript, and we’d like to clarify a few misunderstandings below.
>
>
> >* Weakness 1-1: Reasoning ability of current T2I models
>
> We did not claim that existing visual generative models are not equipped with various reasoning skills. It is a fact that “text-to-image generation (T2I) models suffer from generating objects with specified counts, positions, and attributes” (line 22), which has been discovered in previous work like Stable Diffusion/DALLE/Imagen and etc.
>
>
>
> >* Weakness 1-2: Meaning of discrete categories
>
> We are one of the first work to use LLMs to generate layouts for both 2D and 3D scenes. The sentence refers to previous layout generation models (not LLMs) that are transformer-based and trained from scratch. These models previous a “class” for each bounding box where the class label is represented as a fixed-length one-hot encoding vector instead of free-form text. Thus, these vectors represents “discrete categories”. LLMs is much more flexible that can generate both class labels or detailed descriptions in free-form text.
>
>
>
>
> >* Weakness 1-3: Drawback of existing LLM-centered systems
>
> Line 37-38 refers to previous work like Visual ChatGPT [48] or MM-React [52] that runs image generation models as APIs and directly feed user prompt to the APIs. We will revise the sentence for clarification.
>
>
> >* Weakness 1-4: Motivation of using structured output format during LLM layout planning
>
> We’d like to point out that the all GPT models used in this study (Codex, GPT3.5, GPT4) are trained on both plain text and code snippets, which benefits them to adhere to structured output smoothly. Please find more discussion below in response to your “Weakness 3”.
>
>
> >* Weakness 1-5: Regarding LLM and visual input
>
> We have never claimed that our work enables LLMs to take visual inputs. Our experimental results support the potential because LayoutGPT can understand the spatial concepts (left, right, …) behind coordinate values in 2D or 3D spaces. We also show that LayoutGPT has the potential to handle complicated skeleton sequences with a few examples (Fig. 6 & Table 5 in appendix).
>
>
> >* Weakness 1-6: The meaning of "addressing the inherent multimodal reasoning skills of LLMs"
>
> We focus on designing a method to elicit the visual planning skills from LLMs. Table 3 proves that LLMs cannot effectively use the inherent reasoning skills without structured representations. Fig. 3 in the additional rebuttal PDF materials also substantiate the claim.
>
>
> >* Weakness 2: Regarding notations
>
> Thanks for pointing out, we will unify $o_j$ and $\textbf{o}_j$ in the revision.
>
> $\textbf{o}_j$ refers to the object layout, and our definition of $\textbf{o}_j$ (line 106) includes bbox location ($x_j, y_j$) and bbox size ($w_j, h_j$) through $\textbf{t}_j$ and $\textbf{s}_j$. For 3D objects, $\textbf{o}_j$ also includes orientation $\textbf{r}_j$ (line 108).
>
>
> >* Weakness 3: LayoutGPT with other structured format
>
> In Section 4.4 (page 7, line 201-208), we conducted an ablation study to check the effect of CSS structure and compare it with plain text structure. We compare prompts w/ CSS stuctures (e.g., “teddy bear {width: 32px; height: 45px; left: 31px; top: 9px; }”) and w/o CSS structures (e.g., “teddy bear: 32, 45, 31, 9”). (See Table 2 in the appendix for detailed examples). Results in Table 3 show that wrapping layout w/ CSS structures surpass plain text structure, which verifies the effectiveness of our method.
>
> LayoutGPT’s superior performance w/ CSS structure may due to the fact that OpenAI GPT models have read tons of HTML/CSS code during its pretraining process, and thus have acquired numerical & spatial concepts for webpage planning with similar format.
>
>
>
> >* Weakness 4: GLIGEN’s performance on ground-truth layouts
>
> There are still bottlenecks in existing layout-to-image models. The reason for GLIGEN’s image evaluation scores are two-fold:
>
> (1) GLIGEN has its shortcomings in conducting layout-to-image generation. Even though it is conditioning on groundtruth layout, it may still fail to render images with perfect quality.
>
> (2) The object detection model GLIP also has its shortcomings in detecting objects. This might explain why the generated images have a much lower image-level object accuracy than the layout-level object accuracy.
>
> We mentioned the above reasons in line 186-188, and will include more discussions in revision.
>
>
> >* Weakness 5: Regarding attribute binding
>
> Please see general response.
>
>
> >* Question 1: Providing in-context exemplars in reverse order
>
> We follow previous work [1, 51] for reverse order. The impact of exemplars order is not the main focus of the work.
>
>
> >* Question 2: How are the colored sentences in Figure 4 generated?
>
> We append the following additional paragraph to the instruction with one simple example and 0 ICL exemplars:
>
> *IMPORTANT: apart from generating name and bounding box location for each object, you should also write a detailed description of the object, and add the description to each CSS line. For example, dog {width: 20px; height: 19px; left: 42px; top: 25px; description: a white dog with many black dots}*
>
> >* Question 3: Influence of furniture frequencies in 3D layout planning
>
> We observe that instruction without the furniture frequency has similar KL Div., FID and OOB rates. We append the information for completeness yet finding a proper method to input the distribution remains a future study.
>
>
> >* Question 4: Impact of CSS structure vs. Impact of instruction
>
> The CSS structure explicitly clarifies the spatial meaning of each number with the dense format of “Property name: Property value;”. While instruction also explains the meaning of each value in (x,y,w,h), we hypothesize that it is a weaker implication for LayoutGPT to understand the values.
>
> >* Limitations: Limitation section
>
> We have included a section on LayoutGPT’s limitations in the supplementary material (Section F, line 210-225).

---

> > ### Comment · Reviewer_uKyZ · 2023-08-17
> >
> > Thanks for the response which has addressed most of my concerns.
> > However, I remain uncertain about the concept of attribute binding. Could you provide further explanation on how to understand 'attribute binding' refers to binding attributes to generated grounding box instead of the generated images? Take Fig 4 in the paper for example, if the color of the bounding box of two objects. i.e., 'a brown horse' and 'a white truck', are swapped, I believe that another reasonable image could be generated based on the layout information via GLIGEN. Therefore, it is hard to distinguish between correct and incorrect attribute binging.
> > In addition, regarding the quantitative results on attribute binding, why does A&E achieve better attribute binging performance than LayoutGPT+GLIGEN?
> >
> > Overall, the proposed method effectively explores the capability of LLM in layout planning under 2D images or 3D scenes and has achieved some promising results.

---

> > > ### Author Response · Authors · 2023-08-17
> > > **Official Response by Authors**
> > >
> > > Dear Reviewer uKyZ,
> > >
> > > Thank you for your kind response and follow-up questions. Using Fig.4 as an example, our explanation of “binding attributes to generated grounding box” refers to the outcome that LayoutGPT outputs
> > >
> > > > *“**horse** {width: 40px; height: 40px; left: 12px; top: 12px; description: **a brown horse standing still**}*
> > >
> > > > ***truck** {width: 40px; height: 20px; left: 24px; top: 22px; description: **a white truck with four black wheels**}”*
> > >
> > > instead of
> > >
> > > > *(incorrect, attribute swapped)*
> > >
> > > > *“horse {width: …; description: a **white** horse standing still}*
> > >
> > > > *truck {width: …; description: a **brown** truck with four black wheels}”*
> > >
> > > or
> > >
> > > > *(incorrect, whole description swapped)*
> > >
> > > > *“horse {width: …; description: **a white truck with four black wheels**}*
> > >
> > > > *truck {width: …; description: **a brown horse standing still**}”*
> > >
> > > The word(s) (e.g. horse/truck) ahead of the left curly bracket “{“ defines the category or high-level description of the box. **We interpret the first box as a “horse” box and the second one as a “truck” box.** The “description” property between the curly brackets provides low-level attribute descriptions. **Therefore, LayoutGPT correctly binds the attribute “brown” to the “horse” box, and the attribute “white” to the “truck” box.** As indicated in the general response, LayoutGPT binds attributes to the correct box with 100% accuracy on HRS prompts. We will revise the paper accordingly to avoid further confusion.
> > >
> > > With GLIGEN/ReCo as the downstream model, the system ideally ends up with “a brown horse” and “a white truck” in the images. However, we observe that the description for each box has a weaker influence on the generated object in GLIGEN compared to ReCo. For instance, even though “a brown horse” is provided along with the “horse” box coordinates, GLIGEN fails to generate the correct color more often than ReCo or A&E. **In short, GLIGEN might be weaker than ReCo in controlling local attributes for each object.** We conjecture that the differences originate from the differences in training data: **GLIGEN is trained on boxes associated with short class names without attribute words (e.g. bride, groom in Fig. 2(a) in GLIGEN paper); while ReCo is trained on boxes associated with dense descriptions (e.g. a long white and red bus in Fig. 1(a) in ReCo paper).** As the major bottleneck lies in the downstream models, we believe that the whole framework would be improved with stronger layout-to-image models in the future.
> > >
> > > Please let us know if the explanation is now clear enough. We are always delighted to engage in further discussion and offer responses to your uncertainties. Thank you again for your appreciation of the overall contribution and experimental results of our work.
> > >
> > > Regards,
> > >
> > > Authors of #43

---

> > > > ### Comment · Reviewer_uKyZ · 2023-08-17
> > > >
> > > > Thanks for the detailed response, I have no further questions.

---

### Official Review · Reviewer_HVHZ · 2023-07-07

**Soundness:** 3 good
**Presentation:** 3 good
**Contribution:** 2 fair
**Rating:** 5
**Confidence:** 4

**Summary:**

The paper propose LayoutGPT that can generate visual arrangements of objects using the input prompts, providing a way to collaborate with visual generative models for compositional layout based image generation in both 2D and 3D. Experiment results show that such a method can largely improve layout-based generation using reasoning ability of large language models.

**Strengths:**

1. The paper is well written and easy to follow.
2. The authors have conducted a comprehensive evaluations/ablations on the method.
3. The paper provides an interesting way to connect large language models with visual generative models for image generation without any additional training.

**Weaknesses:**

1. **The paper simply proposes a module using LLMs to conduct visual planning**, i.e., extract / reason object relations and the number of objects when given input text descriptions. For image generation, extracted bounding boxes are simply used as inputs to existing layout-based methods, thus no technical contributions to visual generative models.
2. **The paper over-claimed more or less.** For example, in Line 196, the author claimed "LayoutGPT can perform accurate attribute binding". However, in text-based inpainting section of Figure 4, the purple suitcase doesn't have the specified design "a blue, yellow and white flower" and the cat isn't black and white. Similarly, the spatial relationships extracted from the method seem to be off. The first example of text-based inpainting, the cat should be sitting under a bench, thus it seems to me that the method isn't that reliable.
3. **Lacking some baselines**. For 2D image generation, there are a few works that do layout-based image generation, specifically using bounding boxes. It would be good to compare with these methods. One example I can think of is [1], which also doesn't require additional training.
4. **Related Works**. It would be quite relevant to include compositional image generation, where you generate images conditioned on multiple specifications or objects.

[1] Chen et al., Training-free layout control with cross-attention guidance (CVPR 2023)


**Questions:**

1. In table 4, when generating images, is the mainly prompt used as the input or are those colored sentences used as inputs?
2. It seems to me that layoutGPT can do a lot of guessing instead of reasoning. What if you use counter-factual examples or simple examples that rarely appear in real life?

**Limitations:**

The authors don't include limitation section. One thing is that LLMs can be biased such that generated imagery can also biased.

---

> ### Author Rebuttal · Authors · 2023-08-10
>
> We thank the reviewer for providing suggestions for improvements and would like to clarify a few misunderstandings in the response.
>
> > * Weakness 1: Technical contributions of LayoutGPT
>
> We respectfully disagree that our contribution is limited. We have never claimed that LayoutGPT contributes to visual generative models. Our main contribution is: 1) combining program synthesis with ICL for LLMs to achieve layout generation; 2) NSR-1K benchmark; 3) proving the effectiveness of facilitating visual planning&generation using LLMs. We are also one of the first to show LLMs’ potential in understanding 3D concepts and 3D planning. As far as we know, layout generation alone has been extensively studied in 2D [17, 21, 23, 24] or 3D [31,35,44]. Moreover, LayoutGPT shows strong performance in both domains.
>
>
> > * Weakness 2: The paper over-claimed more or less.
>
> While we respectfully disagree, we do notice the confusion on attribute binding. Please refer to the general response for clarification.
>
> The inaccurate “purple suitcase” and “black and white cat” attributes to the bottleneck from the layout-to-image model. As is claimed above, the downstream layout-to-image model is NOT the main focus of our work. The examples of text-based inpainting aims to shows the flexibility and potential of LayoutGPT in “imagining” intricate object-level language descriptions for creative image generation.  Fig. 1 in additional rebuttal PDF material shows that ReCo [53] can accurately represent each object with output from LayoutGPT.
>
> As for the mistake in spatial relation in Fig. 4 (bottom-left), these examples are generated by using just an instruction and no in-context examplars. We only append the following text to the instruction to enable text-based inpainting:
>
> *IMPORTANT: apart from generating name and bounding box location for each object, you should also write a detailed description of the object, and add the description to each CSS line. For example, dog {{width: 20px; height: 19px; left: 42px; top: 25px; description: a white dog with many black dots}}*
> Hence, the exception in Fig.4 doesn’t represent the overall performance. Please refer to Table 2&3 for systematic evaluation.
>
>
> > * Weakness 3: Lacking baselines like “Training-free layout control with cross-attention guidance (CVPR 2023)”
>
> As is claimed above, our main focus is layout generation, not layout-guided image generation. Yet, in fact, we have included the indicated work upon submission. In Table 3, line 9-10 shows the results of the listed work (which we refer to as “Layout-Guidance”) conditioning on the layouts predicted by LayoutGPT. The results verify that LayoutGPT is a model-agnostic approach, and can be applied to various layout-to-image generation models (line 209-214).
>
>
> > * Weakness 4: Add related work discussion on compositional image generation
>
> We will add a separate subsection for compositional image generation for completeness. We appreciate any specific references you would like to provide.
>
>
> > * Question 1: Input for image generation in Figure 4
>
> For images in Figure 4, we input the main prompt, all the colored sentences, and corresponding bounding boxes to GLIGEN. GLIGEN will condition on the colored sentence when rendering objects for each bounding box. Similarly for ReCo in Fig.1 in the additional rebuttal PDF materials.
>
>
> > * Question 2: It seems to me that layoutGPT can do a lot of guessing instead of reasoning. Counterfactual / rarely seen prompts
>
> We respectfully disagree that LayoutGPT do more guessing than reasoning. We conducted multiple experiments to show the reasoning behind. First, in our appendix, line 98-105 and Table 3 shows that LayoutGPT achieves strong performance with random exemplars. LayoutGPT elicits reasoning abilities instead of copying from exemplars. Second, Fig.2&4 shows that LLMs generates novel sizes/locations instead of copying from the exemplars. Lastly, Table 3 in main paper and Fig. 3 (top) in additional rebuttal PDF materials show that all components are essential for correct spatial relations. Please also kindly refer to the general response.
>
>
> >* Limitations: Limitation section
>
> We would like to clarify an important misunderstanding regarding the limitation section. We have included a limitation section in the supplementary material (Section F, line 210-225) where we have discussed multiple points ranging from layout domains to knowledge distillation. We are happy to include the bias discussion in the next revision.

---

> > ### Comment · Reviewer_HVHZ · 2023-08-18
> >
> > In terms of compositional image generation, it has been many works:
> >
> > For example,
> > 1. Training-Free Structured Diffusion Guidance for Compositional Text-to-Image Synthesis (Feng et al)
> > 2. Exploring Compositional Visual Generation with Latent Classifier Guidance (Shi et al)
> > 3. Compositional Visual Generation with Composable Diffusion Models (Liu et al)
> >
> > thanks the authors for addressing some of my questions. I will keep my rating as it is (borderline accept).

---

> > > ### Author Response · Authors · 2023-08-18
> > > **Thank you for your comments**
> > >
> > > Dear Reviewer HVHZ,
> > >
> > > Thank you for kindly pointing to these related papers. We will add these references in the next revision. Please kindly note that our work mainly focuses on layout planning and generation in 2D&3D spaces rather than downstream image generation methods. While these studies are highly relevant, it may not be suitable to directly compare LayoutGPT with them.
> > >
> > > Meanwhile, please kindly let us know if any of your questions or uncertainties remain unresolved. We are always delighted to engage in further discussion and offer responses to your remaining questions. We sincerely appreciate your active participation.
> > >
> > > Regards,
> > >
> > > Authors of #43

---

> > > > ### Author Response · Authors · 2023-08-19
> > > >
> > > > Dear Reviewer HVHZ,
> > > >
> > > > This is a kind follow-up regarding the previous comments. Since we have addressed "some of your questions", could you please specify the remaining concerns or questions? We are available to address them promptly. However, if there are no further inquiries, we kindly ask you to reconsider the upgrade of your rating.
> > > >
> > > > Thank you for the insights and feedback that you have provided during the review process. Your understanding and collaboration are highly valued.
> > > >
> > > > Regards,
> > > >
> > > > Authors of # 43

---

### Official Review · Reviewer_1i69 · 2023-07-07

**Soundness:** 3 good
**Presentation:** 3 good
**Contribution:** 3 good
**Rating:** 6
**Confidence:** 5

**Summary:**

This paper introduces LayoutGPT, a training-free approach that injects visual commonsense into LLMs and enables generating plausible 2D images and 3D scenes conditioned on layouts based on text conditions. Specifically, the authors experiment with four variants of GPT models: Codex, GPT-3.5, GPT-3.5-chat and GPT-4 and showcase that LLMs can produce meaningful 2D/3D layouts using a CSS (Cascading Style Sheets) format, where every object is modelled as labelled bounding box, parametrized with three random variables indicating its category, size and location. For the case for 3D indoor scene synthesis, they parse the layouts into 3D scenes by simply replacing the bounding boxes with 3D objects from a library of assets. For the case of 2D image synthesis, they rely on GLIGEN that is a layout-to-image model to convert the generated layout to a 2D scene. For the case of 2D image synthesis, the authors compare their model to Stable Diffusion and Attend-and-Excite and evaluate the generated layouts based on the precision, recall and accuracy of the generated bounding boxes. To measure whether the generated image matches the provided text description, the authors report the CLIP/GLIP cosine similarity between text prompts and the generated images. For the task of 3D indoor scene synthesis, the authors compare their model to ATISS and measure the generation quality by reporting the KL-divergence between the object category distributions in the ground-truth and the generated scenes. For both tasks, LayoutGPT outperforms most baselines on most metrics and from the qualitative results, it seems that LayoutGPT faithfully generate layouts that match the input conditioning.

Overall, I think this is a nice work that introduces an elegant way for using LLMs for 2D and 3D layout synthesis. I think that the idea of representing scene layouts in the CSS format is very intuitive and greatly simplifies the task of layout synthesis. My main concern, as also discussed in the Weaknesses section, is related to whether the proposed model can robustly generate (i) layouts with more complex text conditioning for the case of the image synthesis, (ii) 3D scenes conditioned on detailed scene descriptions clearly describing the how many and what objects should be placed in the scene.


**Strengths:**

1. To the best of my knowledge, the idea of using LLM for 2D and 3D layout synthesis is novel and the authors clearly demonstrate that LLMs can produce complex 2D images and 3D scenes in a CSS format. Unlike other concurrent works that try to use LLMs for similar tasks, I think this model is simpler and more generic, hence it can be applied on both 2D scene synthesis and 3D scene synthesis.

2. I particularly liked that the authors provide results on both a 2D and a 3D task. They compare their model with several strong baselines and showcase that the proposed model can consistently produce 2D images/3D that match the input conditioning. Moreover, from the quantitative evaluation, we note that the proposed training-free method achieves state-of-the-art performance w.r.t. most metrics for both tasks.

3. I appreciated the additional ablations as well as the various implementation details provided in the supplementary. In addition, I think that also the proposed NSR-1K benchmark, can potentially be very useful for various tasks. Although the authors don't mention whether they plan on releasing these benchmark, I would like to encourage them to do so as I think such benchmarks can greatly benefit the research community.


**Weaknesses:**

1. For the case of the 3D indoor synthesis task, I am wondering whether the authors tried to condition the scene generation on more detailed text descriptions that go beyond simply specifying the room type and the size of the room. For example, given a description like "a bedroom with one double bed, two nightstands and 1 wardrobe", would LayoutGPT be able to generate a layout that matches this description? I assume that to some extend this should work quite well so I am not sure why the authors did not provide these type of results. I think it would be beneficial for the paper to include them for the final version of the paper.

2. For the experimental evaluation of the 2D scene synthesis, the authors report precision, recall and accuracy. Is there a reason why not also report MeanIoU on the bounding box parameters? I am not sure whether I am missing out something but I think this metric is very important as the model generates bounding boxes in practice. In addition, I think that the section B2 in the supplementary that discusses the Evaluation metrics and in particular the accuracy computation is not very clear. I think it is good to polish this section a bit.

3. Although, I appreciate that the authors proposed a new benchmark for their image synthesis experiment, I think they should have also evaluated their model on the 2017 Panoptic version of COCO dataset that has been previously used by several generative models that perform layout generation. From the description in L152-161, it is not clear to me whether the panoptic version of the COCO dataset is included in the proposed benchmark. Can the authors please clarify this?

**Questions:**

1. I am wondering whether the LLMs can produce diverse layouts conditioned on the same text conditioning. I believe that this is an interesting experiment that the authors should include in their evaluation. In addition, another valuable analysis would be to compare existing GPT models w.r.t. their capabilities for generating diverse layouts conditioned on the same input prompt. Being able to generate diverse layouts is a very important trait of existing models, hence I think the authors should provide additional experiments that demonstrate whether this is possible or not.

2. Looking at the Experimental Evaluation and in particular the Image Synthesis results, I am wondering how robust is the proposed model if the input text prompt contains larger descriptions. Looking at all the results both in the main paper and in the supplement, I think that the authors show layout generations with at most 5 objects (see Fig 3., top row, right column). Have the authors tried to condition the layout generation with more detailed text prompts with more objects? How good would their model work?

3. In Table 1, the authors mention that their proposed NSR-1K benchmark contains several text descriptions that have comparisons e.g. "A picture of three cars with a few fire hydrants, the number of cars
is more than that of fire hydrants". I checked both the main paper and the supplementary but I was not able to find any conditioning like this. I think it would be great if the authors can provide some examples that show that their model works well with these more challenging text conditioning.

4. Can the authors clarify why LayoutGPT cannot work with floor plans of various shapes. In L219-220 of  the main paper they state that it is not compatible with irregular floor plans but I am not sure why this is really an issue?

5. For the quantitative evaluation in Section 5.1 the authors should also mention the image resolution of the rendered images, used to compute the FID scores. Moreover, they mention that to compute the FID score they render scene images from four camera angles (L224-225), are these angles random per scene? I think it is important that the authors clarify this for reproducibility purposes.

6. Some paper references that are missing that I think the authors should include in their final version of their paper are listed below:
* Variational Transformer Networks for Layout Generation, CVPR 2021
* BLT: Bidirectional Layout Transformer for Controllable Layout Generation
* LayoutDM: Discrete Diffusion Model for Controllable Layout Generation, CVPR 2023 (this is a concurrent work but still it might be good to add it in the reference list)

**Limitations:**

The authors discuss the limitations of their work and show several failure cases in their supplementary material. In addition, they also discuss potential negative societal impact of their work.

---

> ### Author Rebuttal · Authors · 2023-08-10
>
> Thank you for your careful reading and insightful comments.
>
> > * Weakness 1: 3D layout planning with detailed description
>
> Thanks for your suggestions. Please note that 3D-FRONT does not have ground truth captions for the rooms. Besides, existing work mostly conditions on room types or floor plans instead of using language descriptions. As a reference, we generate template-based captions (e.g. “bedroom with a double bed, two wardrobes, and a pendant lamp.”) and use them as conditions. As shown in the table below, the near-zero KL Div. value indicates that LayoutGPT faithfully follows the descriptions to generate the type and amount of objects. However, captions do not provide additional information to improve out-of-bounds or FID scores. Please see Fig. 2 in additional materials for visualization.
>
> |                                | Bedroom       |         |       | Living room   |         |       |
> |--------------------------------|---------------|---------|-------|---------------|---------|-------|
> |                                | Out of bounds | KL Div. | FID   | Out of bounds | KL Div. | FID   |
> | LayoutGPT (GPT-3.5, w/caption) | 54.27         | 3.21e-7 | 27.68 | 77.36         | 3.54e-5 | 76.87 |
>
>
> > * Weakness 2: MeanIoU for 2D layout evaluation
>
> We did not use MeanIoU to evaluate layout performance because the nature of the task is generation instead of prediction. There may be inexhaustible possible valid layouts for the same prompt. Besides, the prompt does not describe object sizes or specific locations. Therefore, the MeanIoU between the prediction and the reference image may not adequately evaluate the “correctness” of the prediction in terms of numerical or spatial reasoning.
>
> > * Weakness 2: Metric for 2D layout evaluation
>
> Thanks for your suggestion. We will revise the metric section and provide some visualization examples. The accuracy is basically the percentage of test samples that end up with the correct numerical or spatial relations based on the layout (Layout Acc.) or the detected layout (GLIP Acc.).
>
>
> > * Weakness 3: Regarding COCO2017 panoptic subset
>
> Our NSR-1k benchmark is built on COCO2014 with 80 categories. Thank you for your suggestion and we evaluate LayoutGPT on the Panoptic benchmark. Please see our response to your “Question 2” below for detailed setup and experimental results.
>
>
> > * Question 1: Diversity of the layouts planned by LayoutGPT
>
> Beyond our observation of the diversity, we also generate five different layouts for each prompt and compute the standard deviation (std) of bounding box sizes and locations. The normalized bounding box sizes have a std of +-0.151 and the normalized locations have a std of 0.083. Please also note that users can adjust the diversity by setting different temperatures for the LLMs decoding stage, which further guarantees the diversity of layouts.
>
>
> > * Question 2: Prompting LayoutGPT with more objects
>
> We test LayoutGPT on 500 examples randomly sampled from the validation set of COCO2017 Panoptic with 6~15 annotated bbox (out of the consideration for GPT length limit). We retrieve 8 supportive examples from the train set during in-context learning.
>
> To adapt to the nature of the Panoptic task, we add the following additional instruction when prompting LayoutGPT:
> “The objects layout might be dense, and objects may overlap with each other. Some objects might not have been mentioned in the prompt, but are very likely to appear in the described scenario.”
>
> We generated images using GLIGEN and the LayoutGPT's output and achieve 86 FID score compared to 90 FID using LayoutTransformer[17]+GLIGEN
>
> In Fig.4 of the additional rebuttal PDF material pdf, we show the dense layout predicted by LayoutGPT together with GLIGEN’s visualization. Results show that LayoutGPT can be smoothly applied to more complicated scenarios with more objects (middle: loads of donuts) and with more categories (left: indoor scene; right: outdoor streetview).
>
> It is worth noting that, even though the prompt may only mention a few objects, LayoutGPT is able to predict the layout of the whole scene (as requested in the instruction above), including the objects that may commonly appear under each scenario (e.g., left: towel by the sink, mirror over the sink, vase on the counter, …). This further demonstrates LayoutGPT’s powerful reasoning ability with commonsense knowledge.
>
>
> > * Question 3: Demonstrative examples for numerical reasoning with comparative terms
>
> Thanks for pointing out. We’ll provide more demonstrative examples for numerical reasoning with comparison terms in the next revision due to space constraints. Yet please refer to Fig.4&5 in the additional rebuttal PDF for more than 5 objects per image.
>
>
> > * Question 4: Regarding floor plans with various shapes
>
> Non-rectangular floor plans from 3D-FRONT are represented as a long list of vertices or binary images. At this time, LLMs cannot understand the meaning behind the list or take images as inputs. Therefore, we leave it as future work to improve LLM’s compatibility or multimodal skills.
>
>
> > * Question 5: Image resolution for 3D scene FID
>
> Thanks for pointing out the issue. We render 256x256 images for each scene from four fixed camera positions (0,0,2), (0,2,0), (2,0,0), and (2,2,2) (unit: meters). Cameras always point towards the origin (0,0,0).
>
>
> > * Question 6: Add reference
>
> We will add these references in the next revision for completeness.

---

> > ### Author Response · Authors · 2023-08-20
> > **Official Comment by Authors**
> >
> > As the end of the discussion period is approaching, we are wondering if you have read our rebuttal and if you have any remaining concerns. We are happy to clarify more before the discussion period ends.

---

> > > ### Comment · Reviewer_1i69 · 2023-08-21
> > > **Rebuttal Acknowledgment**
> > >
> > > I would like to thank the reviewers for taking the time to carefully address my questions and concerns. After reading the author's rebuttal and the other reviews, most of my questions are addressed. Although, I am still a bit skeptical regarding how well the proposed model would work if more complex text descriptions would be used as conditioning, I think that even in it's current form it presents an approach that is valuable for the research community, hence I think it should be accepted. Therefore, I would like to keep my original rating.

---

> > > > ### Author Response · Authors · 2023-08-22
> > > > **Thank you for your response**
> > > >
> > > > Dear Reviewer 1i69,
> > > >
> > > > Thank you for your kind response and support.
> > > >
> > > > Regarding more complex text conditions, we have tested LayoutGPT on the following two additional corpus in the rebuttal:
> > > >
> > > > * COCO2017 Panoptic based on your suggestions;
> > > >
> > > > * Rare scenarios or counterfactual prompts generated by ChatGPT (results are included in the General Response and attached PDF).
> > > >
> > > > We would be happy to scale up the experiments and include more discussions on LayoutGPT’s applicable scenes in the next revision. Meanwhile, please feel free to specify any sources of complex text conditions that you’d like to see being tested. If our response addresses your concerns well, we’d be grateful if you would consider increasing the rating. Thank you for your insight and suggestions throughout the reviewing process.
> > > >
> > > > Best regards,
> > > >
> > > > Authors of #43

---

> ### Comment · Area_Chair_v1uq · 2023-08-20
>
> Dear reviewer,
>
> Please look over the author response and the other reviews and update your opinion.  Please ask the authors if you have additional questions before the end of the discussion period.

---

### Official Review · Reviewer_ykMz · 2023-07-10

**Soundness:** 3 good
**Presentation:** 4 excellent
**Contribution:** 3 good
**Rating:** 6
**Confidence:** 5

**Summary:**

This paper proposes LayoutGPT, a method to compose in-context visual demonstrations in style sheet language to enhance the visual planning skills of LLMs. As the first work to use LLMs to generate layouts from text conditions, LayoutGPT can generate plausible layouts for 2D images and 3D indoor scenes, including challenging language concepts like numerical and spatial relations. Those generated layouts can be further used for image generation. When combined with a region-controlled image generation model, LayoutGPT outperforms existing text-to-image generation methods by 20-40% and achieves comparable performance as human users in generating plausible image layouts and obtaining images with the correct object counts or spatial relations.

**Strengths:**

1. This is the first work to explore the ability of LLMs for layout generation. It reveals the spatial reasoning ability of LLMs and might inspire future explorations in this direction.
2. The training-free approach is easy to adopt for various applications.
3. The presentation is clear and easy to follow.

**Weaknesses:**

1. Although this was no such exploration before, the proposed approach is a straightforward application of LLM. It would be better and more inspiring if authors could provide some in-depth analysis of the spatial reasoning abilities of LLM.
2. It is unclear if layoutGPT is robust to the selection of in-context exemplars or not. What's the size of the reference set? What if the text condition describes a rare scenario that does not appear in the reference set? Experiments in the supplementary material show that the performance is sensitive to the number selected in-context exemplars.
3. The evaluation is conducted only on NSR-1K for numerical reasoning and spatial reasoning in text-to-image synthesis and ATISS for indoor scene synthesis. Although the authors claim that layoutGPT can be used for accurate attribute binding and text-based inpainting, only visual results (fig.4) are shown and there are no quantitative experiments on such datasets to demonstrate such abilities and applications.

**Questions:**

Please refer to the questions in the weakness section.

**Limitations:**

The authors mention the limitations in the supplementary materials.

---

> ### Author Rebuttal · Authors · 2023-08-10
>
> Thank you for your valuable feedback.
>
> > * Weakness 1: straightforward application and in-depth analysis of spatial reasoning ability
>
> One of our main contributions is to combine style-sheet program synthesis with LLMs and in-context learning as the CSS style language inherently share similarity with bbox-based planning for visual generation. Different from previous work [48,52] that uses simple representation (x1,y1,x2,y2), our ablation study show that CSS structure elicits much stronger spatial reasoning skills from LLMs (see Table 3). This finding requires the awareness of shared properties between LLMs’ pretraining data (CSS-involved programs) and image/3D layout representations. While “w/ instruction” can achieve considerable spatial accuracy (lines 2&6 in Table 3), CSS structure is more important by comparing lines 6-8, which has been analyzed in Sec. 4.4. Therefore, we respectfully disagree that LayoutGPT is a straightforward application of LLMs.
>
> In addition, we substantiate the claim based on Figure 3 in the attached PDF response. Fig.3 (top) shows that CSS enables LayoutGPT to not only avoid unreasonable overlap but accurately generate spatial relations involving multiple objects. We hypothesize that LLMs understand the integer values in the 2D space much better because CSS explicitly declares the property names for these values.  Fig.3 (bottom) shows the spatial reasoning ability across different versions of GPTs beyond quantitative differences. It’s surprising that GPT-4 can generate the shape and position of “a straw” precisely given that *no straw box examples are provided in the in-context demonstrations*. We hypothesize that either GPT-4 is pre-trained on a more comprehensive collection of layout data or the image branch training benefits layout generation as language tokens.
>
>
> > * Weakness 2: Size of ICL and robustness to exemplar selection
>
> For evaluation results in the main paper, we used eight (k=8) in-context demonstrations for image layouts and eight/four (k=8/4) for 3D bedroom/livingroom synthesis.
>
> In our supplementary Table 3, we show that even with random ICL exemplars or fewer exemplars (k<8), LayoutGPT can achieve comparable layout accuracy and GLIP accuracy for both numerical and spatial prompts. LayoutGPT tends to generate extra objects with random exemplars due to its hallucination nature, yet this does not harm the overall accuracy of counting and spatial reasoning. Besides, “train platform” and “straw” in additional rebuttal PDF material Fig. 3 never appear in the exemplars and are not part of the COCO annotations. However, LayoutGPT still manages to elicit the ability of LLMs to generate their 2D boxes. Therefore, we believe that our method is robust to rare scenarios and various sizes of exemplars.
>
>
> > * Weakness 2: LayoutGPT’s performance on rare scenarios
>
> Please refer to the general response.
>
> > * Weakness 3: Attribute binding & text-based inpainting.
>
> Please refer to the general response for attribute binding clarification and evaluation.
>
> Text-based inpainting is a creative application scenario and has no standard benchmark/evaluation yet. It shows the flexibility and potential of LayoutGPT in “imagining” intricate object-level descriptions for efficient and creative image generation.
>
> > * Weakness 3: More quantitative evaluation of LayoutGPT’s reasoning ability
>
> Please refer to the general response regarding LayoutGPT’s counterfactual prompts. We also report additional quantitative results on LayoutGPT’s size reasoning performance.

---

> ### Comment · Reviewer_ykMz · 2023-08-18
> **Updated review after rebuttal**
>
> Thank authors for the rebuttal. The authors have addressed most of my concerns so I changed my rating to weak accept.

---

> > ### Author Response · Authors · 2023-08-19
> > **Thanks for your response**
> >
> > Dear Reviewer ykMz,
> >
> > Thank you for your kind response and support. We are glad to have addressed most of your concerns. Please let us know if you have additional comments or suggestions.
> >
> > Regards,
> >
> > Authors of #43

---

### Author Rebuttal · Authors · 2023-08-10

# General Response

We thank all reviewers for their constructive feedback and comments. We would like to address reviewers’ common concerns in the following general response:


> * Quantitative and qualitative results regarding attribute binding **(Reviewer ykMz & HVHZ & uKyZ)**

We would like to clarify that “accurate attribute binding” in lines 196 & Fig. 4 refers to binding attributes to generated bounding boxes instead of objects in the generated images. We will revise the writing to avoid future confusion.

Here, we’d like to show more quantitative and qualitative results on attribute binding. LayoutGPT binds attributes to each object’s bounding box with 100\% accuracy on HRS [1] color prompts (e.g. “a green car and a blue chair”). On top of that, we evaluate the attribute correctness rate (accuracy) on the final generated images when LayoutGPT is combined with downstream image generation models. The below indicates that the major bottleneck lies in the layout-guided generation part of the system. Fig.1 in the additional rebuttal PDF material shows that LayoutGPT+ReCo ends up with more faithful object attributes.


|                  | Attribute binding accuracy using HRS eval metric on generated images |                      |                     |                      |
|------------------|----------------------------------------------------------------------|----------------------|---------------------|----------------------|
|                  | Overall                                                              | Prompts w/ 2 objects | Prompt w/ 3 objects | Prompts w/ 4 objects |
| SD1.4            | 12.84                                                                | 18.57                | 10.10               | 11.36                |
| A&E              | 22.96                                                                | 31.43                | 19.19               | 20.45                |
| LayoutGPT+GLIGEN | 18.68                                                                | 22.86                | 19.19               | 14.77                |
| LayoutGPT+ReCo   | **36.96**                                                                | **40.00**                | **37.37**               | **34.09**                |



> * LayoutGPT’s performance on rare scenarios / counterfactual prompts **(Reviewer ykMZ & HVHZ)**

We provide more discussions on LayoutGPT’s performance on rare scenarios or counterfactual prompts.

* Quantitative Results

We first evaluate LayoutGPT’s reasoning ability regarding object size. We use the standard HRS benchmark [1] which is designed for benchmarking compositional text-to-image models.
HRS prompts for size reasoning contain comparison terms between randomly sampled common objects. The size relations described in HRS size prompts are often counterfactual and rarely seen (e.g., “a person which is smaller than a chair and larger than horse”, “a car which is smaller than a banana and chair and bigger than airplane”). LayoutGPT achieves an accuracy of 98.0% / 93.1% / 92.1% when the prompt involves size comparison between 2/3/4 objects. Meanwhile, the best size reasoning performance of 9 text-to-image models reported by the HRS benchmark has only 31.1% / 0.2% / 0%. The results verify that LayoutGPT acquires decent reasoning ability on rare scenarios / counterfactual prompts.


* Qualitative Results

In addition, we ask GPT-4 to write a few counterfactual prompts with the following instructions:

*“Please provide a few counterfactual prompts that depict rarely seen the spatial relationship between the 80 MSCOCO object categories. An example would be "a monkey riding on top of a bird"”.*

We test LayoutGPT on these counterfactual prompts with 8-shot in-context learning. The supportive examples for in-context learning are from the MSCOCO2017 train set that depicts everyday scenarios, which is very different from the ChatGPT-generated counterfactual prompts used for testing.

We show the illustrative demo of LayoutGPT’s prediction in Fig.5 in the additional rebuttal PDF material. LayoutGPT demonstrates competent layout planning ability on these challenging counterfactual prompts and handles the relationship between objects well.


[1] HRS-Bench: Holistic, Reliable and Scalable Benchmark for Text-to-Image Models (ICCV’23).

---

### Author Response · Authors · 2023-08-14
**Welcome Further Response and Discussion**

Dear Reviewers,

Thank you for your valuable review. We have provided responses to your questions, and are committed to addressing further concerns.

We would like to ask for your kind participation in the discussions. Please let us know if we have addressed your concerns or if you have additional feedback or suggestions.

We highly appreciate your time and efforts and are looking forward to the discussions.


Best regards,

Authors of Submission#43

---

### Decision · Program_Chairs · 2023-09-21

**Decision:**

Accept (poster)

**Comment:**

This submission proposes to use large language models (LLMs) with in-context examples to generate layouts with text conditions.  Layouts are represented as css and the method is evaluated on its ability to generate layouts for 2D images (with counting and spatial relations) and 3D indoor scenes (given room type and size).

Overall, reviewers are positive as this is the first work to show that LLMs can be leveraged to generate these layouts with in-context learning without explicit training.  As the work does have limitations, reviewers had additional questions and concerns for the authors which were addressed during the author response period.

The authors are encouraged to incorporate clarifications and reviewer suggestions in their camera ready including:
1. Clarifying details including number of the in-context examples, notation, and other reviewer questions
2. Discussion and experiments on attribute binding.
3. Generation of 3D scenes based on text descriptions that go beyond just the room type and size.
4. Additional examples of what the model can handle well and what it cannot (including rare scenarios and analysis of 2D reasoning ability of the proposed model)
5. Add discussion of compositional image generation to related work
6. Move discussion of limitations to main paper